# Comparisons of ground-based tropospheric NO2 MAX-DOAS measurements to satellite observations with the aid of an air quality model over Thessaloniki area, Greece

Theano Drosoglou[1], Alkiviadis F. Bais[1], Irene Zyrichidou[1], Natalia Kouremeti[1,2], Anastasia Poupkou[1], Natalia Liora[1], Christos Giannaros[1], Maria Elissavet Koukouli[1], Dimitris Balis[1], Dimitrios Melas[1]

[1]Laboratory of Atmospheric Physics, Aristotle University of Thessaloniki, Thessaloniki, 54124, Greece
[2]Physikalisch-Meteorologisches Observatorium Davos, Dorfstrasse 33, CH-7260 Davos Dorf, Switzerland

*Correspondence to*: Theano Drosoglou (tdroso@auth.gr)

**Abstract.** One of the main issues arising from the comparison of ground-based and satellite measurements is the difference in spatial representativeness, which for locations with inhomogeneous spatial distribution of pollutants may lead to significant differences between the two datasets. In order to investigate the spatial variability of tropospheric $NO_2$ within a sub-satellite pixel, a campaign which was lasted for about six months was held at the greater area of Thessaloniki, Greece. Three MAX-DOAS systems performed measurements of tropospheric $NO_2$ columns at different sites representative of urban, sub-urban and rural conditions. The direct comparison of these ground-based measurements with corresponding OMI/Aura products showed good agreement over the rural and sub-urban areas, while the comparison with GOME-2/MetOp-A and GOME2/MetOp-B observations is good only over the rural area. GOME2A and GOME2B sensors show an average underestimation of tropospheric $NO_2$ over the urban area of about $10.51\pm8.32 \times 10^{15}$ and $10.21\pm8.87 \times 10^{15}$ molecules $cm^{-2}$, respectively. The mean difference between ground-based and OMI observations is significantly lower ($6.60\pm5.71 \times 10^{15}$ molecules $cm^{-2}$). The differences found in the comparisons of MAX-DOAS data with the different satellite sensors can be attributed to the higher spatial resolution of OMI, as well as the different overpass times and $NO_2$ retrieval algorithms of the satellites. OMI data were adjusted using factors calculated by an air quality modeling tool, consisting of the Weather Research and Forecasting (WRF) mesoscale meteorological model and the Comprehensive Air Quality Model with Extensions (CAMx) multi-scale photochemical transport model. This approach resulted in significant improvement of the comparisons over the urban monitoring site. The average difference of OMI observations from MAX-DOAS measurements was reduced to -$1.68\pm5.01 \times 10^{15}$ molecules $cm^{-2}$.

Keywords: tropospheric $NO_2$, MAX-DOAS, Phaethon, ground-based, satellite, OMI, GOME-2, air quality modeling, CAMx, Thessaloniki air quality

## 1 Introduction

Nitrogen oxides ($NO_x = NO + NO_2$) are among the most important trace components of the atmosphere playing a key role in the tropospheric photochemistry [e.g. Seinfeld and Pandis, 1998]. They affect the oxidation capacity and the radiative forcing in the lower atmospheric layers by controlling the ozone formation, contributing to nitric acid ($HNO_3$) and nitrate radical
formation and affecting the hydroxyl levels [Solomon et al., 1999; Finlayson-Pitts and Pitts, 2000]. The main emission sources of nitrogen oxides are fossil-fuels combustion, biomass burning, microbiological processes in the soil, lightning and aircrafts [Lee et al., 1997; Jaeglé et al., 2005]. The most important anthropogenic source is the high-temperature combustion processes occurring in vehicle engines and industrial and power plants [EEA report, 2013]. Hence, urban areas, with heavy road traffic, as well as industrial areas are characterized by inhomogeneous spatial and temporal patterns in $NO_x$ concentrations.

Nitrogen dioxide ($NO_2$) is mainly produced by the oxidation of nitrogen monoxide (NO) and only a small proportion of $NO_x$ is emitted directly as $NO_2$ [Hewitt and Jackson, 2009]. However, in the case of diesel vehicles the fraction of directly emitted $NO_2$ to the total $NO_x$ emissions is much higher, resulting in a significant increase of $NO_2$ emissions and more frequent breaching of $NO_2$ air quality limits in urban locations in recent years [Grice et al., 2009; Keuken et al., 2012].

Thessaloniki is the second largest city of Greece and, with a population of more than 1 million inhabitants, hosts about 10%
of the total population of the country [Resident Population Census 2011]. The main air pollution sources in Thessaloniki are road transport, domestic heating and industrial facilities, while the air quality of the city is affected by the local topographic and meteorological characteristics and regional pollution transport [Poupkou et al., 2011]. $NO_2$ concentrations in the city tend to stabilize during the last decades with the highest $NO_2$ levels observed at the traffic hotspots in the center of the city [Moussiopoulos et al., 2008]. According to Zyrichidou et al. [2009] the mean value of tropospheric $NO_2$ column until 2008
over Thessaloniki was $3.9 \pm 3.8 \times 10^{15}$ molecules $cm^{-2}$ and $4.2 \pm 3.8 \times 10^{15}$ molecules $cm^{-2}$ as observed by GOME-2 and OMI satellite instruments correspondingly.

Well established methods are used worldwide for monitoring $NO_2$ concentrations based on both in situ measurements from local air quality networks and remote sensing from ground-based instruments and satellite sensors [e.g. Ordonez et al., 2006; Blond et al., 2007; Brinksma et al., 2008]. Space-borne measurements provide information on $NO_2$ concentrations in a larger
scale and over areas, such as oceans and deserts, where ground-based systems cannot be easily deployed. On the other hand, the spatial and temporal resolution of the satellite data are respectively limited by the satellite footprint size and its overpass time. Thus, well-established, extended and relatively dense ground-based networks in areas where significant spatial and temporal variations in $NO_2$ loading are observed can improve the validation of the satellite data sets.

Remote sensing of $NO_2$ concentrations is based mainly on the Differential Optical Absorption Spectroscopy (DOAS) analysis
[Platt, 1994; Platt and Stutz, 2008] of radiance data, measured either from the space or from the ground. The first studies applying the DOAS method to zenith ground-based measurements for the retrieval of tropospheric and stratospheric $NO_2$ date back to the '70s [Brewer et al., 1973; Noxon et al., 1975]. In the last few decades, the DOAS analysis has been widely applied in ground-based systems for the monitoring of air quality species like $NO_2$ in the atmosphere and is considered as a reference

technique for the validation of satellite observations [e.g. Celarier et al., 2008; Kramer et al., 2008; Chen et al., 2009; Herman et al., 2009; Irie et al., 2012; Li et al., 2013; Ma et al, 2013; Hendrick et al., 2014]. Several studies have shown that satellite sensors underestimate the tropospheric $NO_2$ levels over regions characterized by inhomogeneous pollution loadings such as urban and industrial locations, known also as gradient smoothing effect [Chen et al., 2009; Ma et al., 2013]. Irie et al. [2012] have shown a clear bias of less than 10% between space-borne and ground-based observations over the Tokyo area in Japan, which is characterized by significant spatial variations in $NO_2$ concentrations. In Celarier et al. [2008] the correlation coefficient of the comparison of OMI-derived tropospheric $NO_2$ with different MAX-DOAS instruments at Cabauw, The Netherlands, were found to be about ~0.6. Kramer et al. [2008] estimated similar correlation coefficients between OMI and concurrent MAX-DOAS observations over Leicester, England, while they found significant underestimation of OMI in late-autumn and winter months when comparing with weighted near-surface concentrations. Ma et al. [2013] have shown a systematic underestimation of about 26-38 % in tropospheric $NO_2$ over Beijing by OMI, depending on the satellite retrieval algorithm and time period.

In this study, tropospheric $NO_2$ column measurements derived from satellite sensors (OMI/Aura, hereafter OMI, GOME-2/MetOp-A, hereafter GOME2A, and GOME-2/MetOp-B, hereafter GOME2B) are compared with data from MAX-DOAS systems that were deployed in three different sites within the greater area of Thessaloniki, Greece, for a period of a few months. Air quality at these locations, is representative of urban, sub-urban and rural conditions. In order to minimize the effect of smoothing of $NO_2$ gradients in the satellite data and improve their comparison with the ground based data, adjustment factors (AF) were calculated by an air quality modeling tool, consisting of the WRF meteorological model and the CAMx air quality model, and were applied to OMI data.

## 2 Instrumentation and data

### 2.1 The Phaethon system

Phaethon is a miniature ground-based MAX-DOAS system that performs fast spectrally resolved measurements in the wavelength range 300-450 nm which are used for the retrieval of total and tropospheric columns of atmospheric trace gases. The prototype system (Phaethon #1) was developed in 2006 at the Laboratory of Atmospheric Physics (LAP) in Thessaloniki, Greece [Kouremeti et al., 2008; Kouremeti et al., 2013]. Phaethon has been recently upgraded to improve its performance and two new clone systems (Phaethon #2 and #3) have been assembled. The system comprises a cooled miniature CCD spectrograph (AvaSpec-ULS2048LTEC) by Avantes (http://www.avantes.com/), the entrance optics and a 2-axis tracker. The spectrometer is a symmetrical Czerny-Turner type with 75 mm focal length and a grating of 1800 lines/mm. The slit function has been measured using Cd and Hg spectral lamps and the tunable laser system PLACOS [Nevas et al., 2014] and was found very similar to Gaussian with a full width at half maximum (FWHM) of about 0.25nm for Phaethon #1 and ~0.4 nm for Phaethon #2 and #3. The CCD detector is a Sony 2048-pixel linear array with Deep UV coating that enhances its response below 350 nm and is thermoelectrically cooled to 5°C. The entrance optics comprises a telescope with a planoconvex

lens which focusses the collected radiation onto one end of an optical fiber, with a field of view of about 1.5°. Neutral-density optical filters, cut-off filters and transmission diffuser plates, alone or in various combinations are placed on a filter wheel with 8 positions. One position is clear for scattered radiation measurements and one is opaque for dark-signal measurements. The collected light is transferred through a fused silica UV/Vis optical fiber with high concentration of hydroxyl (OH) and a numerical aperture of 0.22 to the spectrograph entrance slit. The entrance optics is mounted on a 2-axes tracker with pointing resolution of 0.125°, allowing accurate tracking for both direct-sun and sky-radiance measurements at different elevation and azimuth angles. The operating software controls the positioning of the tracker and filter wheel, as well as the data acquisition.

## 2.2 Ground-based measurements

In the framework of the "Optimization and expansion of ground infrastructure for the validation of satellite-derived column densities of atmospheric species" (AVANTI) project, a short campaign was organized to investigate the spatial variability of tropospheric columns of air pollutants in the greater area of Thessaloniki and the effect of this variability on the comparisons of satellite with ground based data. In addition to the Phaethon #1 system that operates regularly at the roof of the Physics Department in the Aristotle University campus which is located in the center of Thessaloniki with prevailing urban conditions (UC), two identical systems (Phaethon #2 and #3) have been deployed at two different locations within an area of about 13 km by 28 km (see Fig. 1 and Table 1). These three locations are characterized by diverse local atmospheric pollution patterns representing urban, suburban and rural conditions. Phaethon #2 was installed at a site with rural conditions (RC) near the sea shore about 28 km south of Thessaloniki city center, where it was operating from November 1$^{st}$ 2014 through January 31$^{st}$ 2015. Phaethon #3 was installed at the Alexander Technological Educational Institute of Thessaloniki (ATEITH) located in an area with sub-urban conditions (SC) ~13 km north-west of Thessaloniki center and it has been operating there from January 20$^{th}$ to May 11$^{th}$ 2015. Unfortunately, due to a technical problem in one of the spectrometers only a short period of parallel measurements is available at all three locations.

During the campaign the MAX-DOAS systems were performing both direct-sun and scattered light measurements. A sequence of sky radiance measurements included the zenith direction and the off-axis elevation angles 2°, 3°, 4°, 5°, 8°, 10°, 12°, 15°, 30° and 45°. Elevation sequences were performed at azimuth angles of 80° relative to the solar azimuth. Additionally, the same sequence of elevation angles was repeated several times during the day at a fixed azimuth angle of 255°, a direction free of significant obstacles in case of UC site. For this study we used observations at all available azimuth angles, considering that no significant variations with azimuth angle were observed, but only at the elevation angles of 15° and 30° in order to avoid uncertainties introduced due to aerosols at lower elevation angles [Hönninger et al, 2004]. An average of measurements at both elevation angles was calculated if the derived VCDs agreed within 20%, otherwise only data for 30° were compared with satellite retrievals. A similar approach has been applied by Brinksma et al. [2008].

The measured sky radiance spectra were analyzed by means of the QDOAS v2.109_3 software developed by the Royal Belgian Institute for Space Aeronomy (BIRA-IASB) and S[&]T (https://www.stcorp.nl/) [Danckaert et al., 2015]. For the DOAS analysis the fitting window 400-450 nm was used. A fourth-order polynomial, a second-order offset and a Ring effect spectrum,

calculated according to the approach described by Chance and Spurr [1997], were also included in the DOAS analysis. Along with the $NO_2$ cross section at 298K [Vandaele et al., 1998], the cross sections of $O_3$ at 223K [Bogumil et al., 2003], $O_4$ at 296K [Greenblatt et al., 1990] and $H_2O$ (HITRAN database, https://www.cfa.harvard.edu/hitran/) were also taken into account. Fig. 2 presents an example of $NO_2$ slant column fitting for an elevation angle of 5º obtained for the UC site on 6 November 2014,

around 10:30 UTC (SZA about 57º). For the retrieval of tropospheric $NO_2$ the zenith spectrum of each sequence of elevation angles was used as reference in order to minimize the influence of the stratospheric component in the calculated off-axis differential slant column densities (dSCDs) [Hönninger et al, 2004]. The tropospheric vertical column density (VCD) can be calculated for each elevation angle by the formula suggested in Ma et al. [2013]:

$$VCD_{trop} = \frac{dSCD_{trop}}{dAMF_{trop}} \qquad (1)$$

where $dAMF_{trop}$ is the tropospheric differential air-mass factor which represents the absorption enhancement in the effective light path caused by the combination of single and multiple scattering and absorption processes.

Look up tables (LUTs) were constructed using radiative transfer simulations to derive the tropospheric $NO_2$ AMF for each measurement location. These simulations were performed by means of the modeling package libRadtran version 1.7 [Mayer and Kylling, 2005] using a pseudo-spherical discrete ordinate radiative transfer method [Buras et al, 2011]. Typical values of

aerosol single scattering albedo (0.95), aerosol asymmetry factor (0.7) and surface albedo (0.1 for UC and 0.07 for the other two sites) were assumed [Bais et al., 2005; Kazantzidis et al., 2006]. A key parameter significantly affecting the radiative transfer simulations and therefore the calculation of AMFs is the trace gas vertical distribution in the atmosphere. In this study, mean vertical profiles were provided by the air quality modeling tool consisting of the photochemical grid model CAMx and the mesoscale weather prediction system WRF. The description of this modeling tool and details about the simulations are

presented in the next section. These vertical profiles decay approximately exponentially with altitude with a scale height of about 0.3 km for UC, 0.65 km for SC and 0.66 km for RC sites. The profiles of SC and RC are comparable, but the $NO_2$ concentrations in the UC profile are much higher, up to about seven times higher in the lowest atmospheric layer near the surface. For the AMF LUTs the mean vertical profile of the aerosol optical depth (AOD) at 355 nm calculated from observations of the LIDAR system operating in LAP/AUTH during the period 2001-2007 [Giannakaki et al., 2010] was scaled

by AOD values in the range 0-1.5. This aerosol information was used for all three locations since, according to Kazadzis et al. [2009], the spatial variability in AOD and its vertical profile in these locations is quite small. In that study, the average differences of AOD at 340 nm at the RC and SC sites from the UC site were found to be -0.07 and -0.01, respectively, for AOD values ranging between about 0.1 and 1.1, while the long-term (1997-2005) mean AOD over Thessaloniki (UC) is 0.33±0.14 and 0.53±0.17 for winter and summertime, respectively [Kazadzis et al., 2007]. Along with the AOD, other variables

considered for the construction of LUTs are: the solar zenith angle, the elevation viewing angle and the azimuth angle relative to the solar azimuth. An example of the AMFs calculated at elevation angles 15º and 30º versus AOD for each location is presented in Fig. 3. The AMF corresponding to each measurement is calculated by multi-linear interpolation, using AOD measurements from the CIMEL sun-photometer operating at Thessaloniki (http://aeronet.gsfc.nasa.gov/).

Prior to their deployment at the campaign sites, the three systems were operating for a few days in parallel in the University campus and their inter-comparison tests revealed a very good agreement and no systematic differences (Fig. 4, Table 2). The hourly mean tropospheric $NO_2$ measurements performed at both 15˚ and 30˚ elevation angles were included in the inter-comparison.

**2.3 Satellite observations**

The Ozone Monitoring Instrument (OMI) is one of four instruments on board the NASA EOS-Aura spacecraft (http://aura.gsfc.nasa.gov/), launched on 15 July 2004 in a sun-synchronous ascending near-polar orbit with around 1:45 pm local equator crossing time [Levelt et al., 2006]. The OMI detector is a 2-D charge-coupled device (CCD) array. OMI is a compact nadir viewing, wide swath (of 2600 km that permits a near daily global coverage), ultraviolet-visible (270 nm to 500

nm) imaging spectrometer that was contributed to the Aura mission by the Netherlands and Finland. The foot pixel size of OMI at nadir is 13 km × 24 km, degrading towards swath edges (up to 40 km × 160 km at the two ends of the track) [Wenig et al., 2008; Curci et al., 2010]. Beginning in 2007, the so-called "row anomaly" [KNMI, 2012] affected some of the cross-track positions of the swath, reducing the spatial coverage of the instrument. In this work, we use only the unaffected pixels. OMI $NO_2$ retrievals are obtained from the spectral measurements in the visible spectral range, between 405 and 465 nm.

Overpass data of the next-generation version 3 (V3) of the OMI $NO_2$ standard product (SP), based on the sequential DOAS fitting algorithm [Marchenko et al., 2015], are used in this study. The algorithm used to create $NO_2$ SP is described by Bucsela et al. [2006], Celarier et al. [2008], Wenig et al. [2008] and Bucsela et al. [2013]. Marchenko et al. [2015] proposed revisions of the spectral fitting in the OMI $NO_2$ retrieval algorithm, which reduce the slant column densities by 10–35 %, bringing them closer to independent measurements. The new V3 algorithm also includes improved resolution (1º latitude and 1.25º longitude)

a priori $NO_2$ profiles from the Global Modeling Initiative chemistry-transport model with yearly varying emissions. The present version uses annual monthly profiles from 2004 to 2014. For dates starting in 2015, the 2014 monthly profiles are used. The use of monthly $NO_2$ profile shapes captures the seasonal variation in $NO_2$ profiles [Lamsal et al., 2010]. The a priori profiles' shape presents a close-to-exponential decay with altitude. These profiles are comparable to the mean CAMx profiles that were used as a priori for the SC and RC sites, and, compared to the profile for UC site, they contain much lower $NO_2$ concentrations

in the lowest atmospheric layers [see also section 2.2]. The description and the improvements of the new V3 $NO_2$ SP are given in Bucsela et al. [2016] and the references therein in details. The operational total and tropospheric $NO_2$ columns from OMI are generated by NASA and distributed by the Aura Validation Data Center (AVDC) (http://avdc.gsfc.nasa.gov).

The Global Ozone Monitoring Experiment-2 (GOME-2) instrument is a nadir-viewing scanning spectrometer that samples the 240–790 nm spectral range with a spectral resolution between 0.24–0.5 nm, with an across-track scan time of 6 seconds and a

default swath width of 1920 km [Callies et al., 2000]. Currently there are two GOME-2 instruments operating, one on board EUMETSAT's Meteorological Operational Satellite -A (MetOp-A), launched in October 2006, and the other mounted on the MetOp-B satellite, launched in September 2012. GOME-2 ground pixels have a default footprint size of 80 km x 40 km. Since July 2013, in the tandem mode, GOME2A operates on a reduced swath width of 960 km with an increased spatial resolution

(approximately 40 km × 40 km), while GOME2B operates on a nominal wide swath at 1920 km. MetOp-A and MetOp-B are flying on a sun-synchronous orbit with an equator crossing time of 09:30 local time (descending node) and a repeat cycle of 29 days. Global coverage of the sunlit part of the atmosphere can be achieved almost within 1.5 days. From the start of GOME-2 measurements in 2007, the instrument has suffered from sensitivity degradation [Dikty and Richter, 2011; Azam et al., 2015],

especially for the wavelengths between 300 and 450 nm (likely due to contamination of optical surfaces). The degradation rates for GOME2B are similar to those for GOME2A (see http://www.eumetsat.int/website/home/TechnicalBulletins/ GOME2/index.html). However, the fitting residuals of GOME2B compared to those of GOME2A are much smaller in 2013 and higher in early 2007 [Hao et al., 2014]. According to Hassinen et al. [2016], the impact of GOME-2 instrument degradation on Level-2 product quality depends on the spectral region and the type of retrieval methods selected.

The operational GOME-2 total and tropospheric $NO_2$ columns from MetOp-A and MetOp-B are generated by the German Aerospace Center (DLR) using the UPAS (Universal Processor for UV/VIS Atmospheric Spectrometers) environment version 1.3.9, implementing the Level-1-to-2 GOME Data Processor (GDP) version 4.7 algorithm (http://atmos.eoc.dlr.de/gome2/). GDP 4.7 is based on DOAS-style algorithms, originally developed for GOME/ERS-2 [e.g. Spurr et al., 2004]. The data processing is commissioned by EUMETSAT within the auspices of the Satellite Application Facility for Atmospheric

composition and UV radiation, O3MSAF, project. The algorithm has two major steps: a DOAS least-squares fitting for the trace gas SCD, followed by the computation of a suitable AMF for the conversion to the VCD. Total $NO_2$ columns, including a tropospheric and stratospheric component, are retrieved with the DOAS method in the visible wavelength range 425–450 nm [Valks et al., 2011]. In DOAS fitting for optically thin absorbers, such as $NO_2$ in the visible, the basic model is the Beer-Lambert extinction law. The a priori $NO_2$ profiles are obtained from a run of the global chemistry transport model (CTM)

MOZART version 2 (Horowitz et al., 2003). The profiles can be fitted by an exponential decrease and, similarly to OMI a priori profiles, compare better with RC and SC sites. More details on the GDP 4.7 algorithm can be found in Valks et al. [2011] and Hassinen et al. [2016].

The main features of the two data algorithms are summarized in Table 3. Both satellite data sets show generally good agreement with independent $NO_2$ measurements [e.g. Valks et al., 2011; Ialongo et al., 2016]. Valks et al. [2011] showed that pollution

episodes over the Observatoire de Haute Provence (OHP) were well captured by GOME-2 and the monthly mean tropospheric columns of MAX-DOAS and GOME-2 are in very good agreement, with differences generally within $0.5 \times 10^{15}$ molecules/cm$^2$. In Ialongo et al. [2016] a moderate correlation (r=0.51) was found between the $NO_2$ total columns' measurements performed by the Pandora spectrometer and the OMI $NO_2$ V3 in Helsinki 2012. Pandora overestimated the SP V3 and the median relative difference was 32±18%.

For each of the campaign locations an overpass data set was extracted from OMI, GOME2A and GOME2B observations. Details of this data selection are discussed in Sect. 3.2. In Fig. 5 the average tropospheric $NO_2$ spatial distribution observed by OMI, GOME2A and GOME2B during the campaign period is illustrated. The maps cover a large area around the three campaign stations to account for the satellite sensors' coarse spatial resolution. The many white cells, which correspond to

lack of sufficiently good quality data, on the OMI map result from OMI sensor's relatively high spatial resolution and row anomalies.

## 2.4 Air quality simulations

The comparison of satellite-derived tropospheric $NO_2$ with ground-based observations in areas with inhomogeneous distribution of air pollution is usually poor, due to the different geometries associated with the measurement methods of the two datasets [Celarier et al., 2008; Kramer et al., 2008; Irie et al., 2012]. Ground-based measurements are representative of the absorption of radiation in a particular viewing direction and path, while measurements of satellite sensors are sensitive to absorption of radiances emerging from a wide area determined by the size of the satellite pixel. In order to overcome this problem and improve the comparisons in the greater area of Thessaloniki, OMI data are adjusted using factors derived from air quality simulations.

The modeling system employed for the calculation of the AFs consisted of the Comprehensive Air Quality Model with Extensions (CAMx, version 5.3) [ENVIRON, 2010] off-line coupled with the Weather Research and Forecasting - Advanced Research Weather (WRF - ARW, version 3.5.1) [Skamarock et al., 2008]. The model simulations were performed for the period November 2014 – May 2015.

The WRF - ARW is a next-generation mesoscale Numerical Weather Prediction (NWP) model [Kalnay E., 2003] designed to serve both atmospheric research needs and operational weather forecasting. WRF simulations were carried out over two domains in Lambert Conic Conformal projection; a 6 km resolution coarse domain (d01, mesh size of 343 × 273) that covers the greater area of Balkan Peninsula and a nested (two-way nesting) domain (d02) that focuses over the area of Thessaloniki with a higher spatial resolution of 2 km (mesh size of 60 × 60). The domains' vertical profile extends up to 16 km above ground level and contains 28 layers of varying thickness with higher resolution near the ground. The initial and boundary meteorological conditions were taken from the European Centre for Medium – Range Weather Forecast, ECMWF, in spatial resolution of 0.125° × 0.125° (~12.5 km) and temporal resolution of 6 h. Microphysical processes were parameterized using the New Thompson et al. [2008] scheme, whereas convection in the coarse domain (d01) was parameterized with the Grell-Devenyi (GD) ensemble scheme [Grell and Devenyi, 2002]. For the innermost domain (d02), no convective parameterization was used. Radiative transfer processes were handled with the shortwave and longwave RRTMG schemes [Iacono et al., 2008]. The surface layer was parameterized using the Eta similarity scheme [Janjic, 2002], while for the planetary boundary layer the Mellor-Yamada-Janjic parameterization scheme [Janjic and Zavisa, 1994] was used. Finally, for the parameterization of land surface processes the Noah Land Surface Model [Tewari, et al., 2004] was applied.

CAMx is a 3D Eulerian photochemical dispersion model widely used in air quality studies during the last decade [Lei et al., 2007; Lee et al., 2009; Kukkonen et al., 2012; Poukou et al., 2014; Liora et al., 2016] several of which are related to joint analysis of simulated and remotely sensed pollutant concentration data [Zyrichidou et al 2009; 2013; 2015; Zyryanov et al 2012; Huijnen et al 2010]. In the current study, the simulation domains of CAMx were identical, in terms of horizontal spatial resolution and projection, with those of the WRF model in order to avoid errors introduced by interpolations between grids,

but were slightly less spatially extended (337 x 267 cells for the Balkan domain and $56 \times 56$ cells for the Thessaloniki domain). CAMx grids were structured in 22 vertical layers extending up to 10 km above ground level. The gaseous and particulate matter anthropogenic emissions for the Balkan domain were derived from the European scale emission database of The Netherlands Organisation (TNO) for the reference year 2007 [Kuenen et al., 2011]. For the greater area of Thessaloniki, the anthropogenic emissions were calculated using mainly the methodologies and emission factors of the EMEP/CORINAIR emission inventory guidebook [EEA, 2006]. More specifically, the anthropogenic emission model MOSESS (Model for the Spatial and Temporal Distribution of Emissions) [Markakis et al., 2013] was applied for the calculation of CO, NOx, $SO_2$, $NH_3$, NMVOC, PM10 and PM2.5 emissions as well as for their chemical, spatial and temporal analysis. The total NOx emissions for the domain of Thessaloniki for the months November to May are presented in the upper panel of Fig. 6. The monthly NOx emissions pattern for each of the months November to May is similar to that presented in Fig. 6 characterized by higher emission density over the urban center due to enhanced anthropogenic activities (mostly road transport). Particulate matter emissions from natural sources (windblown dust, sea salt) as well as biogenic volatile organic compounds from vegetation were estimated using the Natural Emissions model (NEMO) [Poupkou et al., 2010; Liora et al., 2015; Liora et al., 2016]. The gas-phase chemical mechanism employed in CAMx was the 2005 version of Carbon Bond (CB05) [Yarwood et al., 2005]. The chemical boundary conditions for CAMx runs were taken from the global model system C-IFS-TM5 results, available in the framework of the EU project MACC-III.

The adjustment factors of OMI data were calculated by the following procedure: For each 2 km $\times$ 2 km grid cell of the simulations domain the tropospheric $NO_2$ VCD was derived by integrating vertically the model-derived hourly mean mixing ratios of $NO_2$. The lower panel in Fig. 6 presents the simulated $NO_2$ VCD averaged over the period of the campaign for each cell of the simulations domain. It can be clearly seen that the tropospheric $NO_2$ follows the NOx emissions pattern. Then the tropospheric $NO_2$ VCD for the grid cell that contains the coordinates of each monitoring site was divided by the simulated VCD averaged over an area that corresponds to the OMI pixel. Finally, the OMI data were multiplied by these AFs in order to minimize the effect of the differences in the spatial distribution of $NO_2$ and achieve better agreement with the ground-based observations, which are assumed to represent mostly the VCD above the grid cell where they are located.

In reality, the size of the OMI pixels is not always equal to the typical nadir pixel [de Graaf et al., 2016]. For this reason, the average value of the simulated tropospheric $NO_2$ VCD over the area covered by the OMI pixel was calculated using a close-to-actual pixel size and position for each overpass. The pixel is assumed to be centered with the CAMx grid cell containing the cross-track position (CTP) coordinates and its size is estimated from the CTP number according to the OMI Data User's Guide [2012].

It should be mentioned here that the method was applied only to OMI data because GOME-2 pixel is very large, covering typically about one third, in case of GOME2A, or about half, in case of GOME2B, of the entire domain shown in Fig. 6. A similar method has been previously employed on comparisons of GOME2A tropospheric $NO_2$ observations with ground-based data over the Observatoire de Haute Provence (OHP) at southern France [Lambert et al., 2011], leading to a slight improvement of the correlation coefficient from 0.63 to 0.71 and the slope of the linear regression from 0.69 to 0.73.

It should be also noted that, based on calculations of the representativeness area of the MAX-DOAS measurements presented in this study, the 2 km x 2 km model cell used for the calculation of AFs can be considered representative of the measurement light path. For MAX-DOAS observations near the surface under high aerosol conditions an approach based on $O_4$ retrieval should be used for the estimation of the horizontal representativeness area [Wagner et al., 2004; Richter et al., 2013]. However,

the effect of the aerosol loading and, thus, the atmospheric visibility for observations at the elevation angles of 15˚ and 30˚ can be considered relatively weak. Moreover, the horizontal distance, of which these measurements are representative, is quite small, especially for an absorber layer located near the surface. Hence, the horizontal sensitivity range was estimated for the campaign period and OMI overpass time using the elevation viewing angle and boundary layer height (BLH) reanalysis data from ECMWF, as the ratio of the BLH to the tangent of the elevation angle (Thomas Wagner, personal communication). By

this method, an average value of 0.55 km was obtained. During spring the representativeness area can be up to about 10 km, whereas in summer even larger distances can be reached. However, the horizontal distance is larger than 2 km for only about 2% of the campaign data, because the majority of the measurements were performed from late-autumn through early-spring, when the BLH is shallower. Irie et al. [2011] estimated a horizontal sensitivity distance of roughly 10 km for MAX-DOAS measurements performed at Cabauw, the Netherlands during the period from 8 June to 24 July 2009 using a geometrical

approach based on box-AMFs.

## 3 Results and discussion

### 3.1 NO$_2$ tropospheric columns in the greater area of Thessaloniki

The hourly mean tropospheric NO$_2$ measurements obtained by Phaethon at the three campaign sites are presented in Fig. 7. The NO$_2$ at the city center of Thessaloniki (UC) in blue circles are much higher than at the SC (red circles) and RC (yellow

circles) sites, especially during winter months. The NO$_2$ levels observed in the urban area during spring months are slightly lower compared to those in the winter considering also the small temporal variability of monthly road transport emissions, representing the major emission source in the urban center. The averaged measured tropospheric NO$_2$ over the campaign period is ~12.32±7.46 × $10^{15}$ molecules cm$^{-2}$ for the center of Thessaloniki, whereas the mean values for the SC and RC sites are 5.94±3.58 × $10^{15}$ and 4.79±2.36 × $10^{15}$ molecules cm$^{-2}$, respectively. The average tropospheric NO$_2$ at SC and RC sites are in

general comparable. However, some positive excursions are observed at SC because this site is likely affected by NO$_2$ transported from urban areas or by local pollution sources depending on weather conditions. The SC site is adjacent to the city center and to Thessaloniki's industrial area, although industrial activity has been drastically reduced over the last five years. Negative values of the VCD obtained from measurements at higher elevation angles, such those used in this study (15˚ and 30˚ as discussed in section 2.2), indicate that the NO$_2$ absorption of the Fraunhofer reference spectrum is underestimated and

cannot be assumed negligible compared to the analyzed spectra.

## 3.2 Comparison of ground-based tropospheric NO₂ with OMI, GOME-2/MetOp-A and GOME-2/MetOp-B observations

For each of the three campaign sites ground-based measurements of tropospheric $NO_2$ (both 15° and 30° were used as described in section 2.2) are compared with products from satellite overpass data. The collocation criteria used are the solar zenith angle (SZA), the cloud fraction (CLF) and the temporal and spatial difference of the measurements. More specifically, only satellite data corresponding to SZA ≤ 75° and CLF ≤ 20% (observations with radiance reflectance from clouds of less than 50% [van der A et al., 2008]) were selected and averages of Phaethon measurements within ±30 min around the overpass time were included in the comparison. Moreover, OMI pixels with CTP between 7 and 54 (corresponding to cross-track dimension smaller than 60km) and only those that are not affected by the OMI row anomalies [see OMI Data User's Guide, 2012] were used in this study in order to obtain good quality and meaningful data. Finally, the upper limit for the distance between the measurement site and the center of the sub-satellite pixel was set to 25 km for OMI and 50 km for GOME-2 sensors, in correspondence to their typical pixel sizes. In the case of OMI, taking into account the smaller pixel size, the closest pixel was selected for the comparisons, whereas in the case of GOME-2 sensors the average measurement of all pixels within the distance limit of 50 km was calculated. However, the comparison results seem not to be significantly affected when the 50 km limit is used for both OMI and GOME-2 sensors, or when the closest GOME-2 pixel is chosen instead of the average value.

Scatter plots of tropospheric $NO_2$ data for each campaign site and satellite sensor separately are presented in Fig. 8, while averages and statistics of the comparisons are shown respectively in Tables 4 and 5. The $NO_2$ time series over the UC site has been separated into two subsets which coincide with the different periods of measurements at the RC and SC sites, i.e. from 1 November 2014 through 31 January 2015 and from 20 January through May 2015 respectively. The satellite instruments seem to underestimate significantly the $NO_2$ levels over the urban area of Thessaloniki (right column in Fig. 8, Table 4 and Table 5). This finding might be ascribed to the fact that the satellite-derived columns represent the average pollution loading in the sub-satellite pixel area (gradient smoothing effect), while ground-based data are mostly representative of the $NO_2$ amounts in air masses above and in close proximity to the monitoring site. In the case of Thessaloniki, the strong $NO_2$ gradients with significant contribution of lower pollution levels in the suburbs result into lower estimates of $NO_2$ by the satellite. Several studies have reported significant differences in air pollution between the city center and the suburbs using long-term observations by the air quality monitoring network of Central Macedonia [Poupkou et al., 2011; Moussiopoulos et al., 2008]. The statistics calculated from the comparison between the ground-based and OMI measurements over the UC and SC stations (Table 5) are similar to those found in previous studies. For example, the correlation coefficients estimated by Kramer et al. [2008] and Celarier et al. [2008] at Leicester, England and Cabauw, The Netherlands, respectively, are about 0.6. In Ialongo et al. [2016] linear fit slopes of 0.49 and 0.39 for V2.1 and V3 OMI retrieval algorithms, respectively, and r = 0.51 for both were found when compared with Pandora observations in Helsinki. Similarly, OMI measurements at NASA Goddard Space Flight Center were ~25% lower than the Brewer data with an r value of 0.58 in daily values, a slope of 0.75 (±0.14) and an intercept of -0.38 (±2.5) × $10^{15}$ molecules/cm² [Wenig et al., 2008].

The fact that OMI observations compare better with the ground-based $NO_2$ data than the GOME-2 sensors can be attributed mainly to the different satellite pixel sizes and to OMI's high sensitivity in the boundary layer [e.g. Wallace et al., 2009]. The OMI sensor provides better detection of higher local pollution levels [Kramer et al., 2008], due to its higher spatial resolution (13 km × 24 km at nadir), compared to GOME-2 sensors which detect the average $NO_2$ concentration over a larger area (40 km × 40 km for MetOpA and 80 km × 40 km for MetOpB). Moreover, the differences observed between the OMI and GOME-2 data sets can be explained by the difference in satellite overpass time. OMI passes over Thessaloniki at around local solar noon (~10:30 UT), when tropospheric $NO_2$ levels are normally reduced due to photochemical processes [Crutzen, 1979]. On the contrary, GOME2A and GOME2B pass over Thessaloniki during morning hours (between 7:30 - 9:00 UT), when local $NO_2$ concentrations in the city center are usually much higher. The different overpass times may also explain partly the larger deviations of GOME-2 from MAX-DOAS compared to those of OMI. The ground-based observations near the OMI overpass time are lower and, thus, closer to the spatially averaged satellite measurements. The different algorithms used for the retrieval of tropospheric $NO_2$ from OMI and GOME-2 measurements, and mainly the different spectral ranges used in the DOAS analysis and different a priori profiles (Table 3), also play an important role.

The r values calculated for GOME2A and GOME2B over UC site for the whole campaign period are 0.28 and 0.19 respectively. Better correlation coefficient and slope values, 0.51 and 1.2 (±0.049) respectively, between daily averaged MAX-DOAS and GOME2A tropospheric $NO_2$ columns were found by Valks et al. [2011], showing that pollution episodes over the Observatoire de Haute Provence (OHP) were well captured by GOME-2. Ground-based and satellite-derived $NO_2$ tropospheric VCDs are in better agreement over the rural (RC) and sub-urban (SC) stations (left and middle columns respectively in Fig. 8, Table 4 and Table 5). However, there is poor correlation between Phaethon and GOME-2 sensors over those two locations (Table 5) probably due to the limited number of data pairs and some elevated $NO_2$ concentrations observed by the Phaethon system over the SC location which are not detected by GOME-2 sensors. Differences observed between GOME2A and GOME2B results (Fig. 8, Tables 4 and 5) could be attributed mainly to their different pixel sizes, as GOME2B footprint has double the size of GOME2A, and partly to their different instrumental degradation, considering that the same retrieval algorithm is used. In general, the slope, intercept and r values from the comparison between GOME2A and MAX-DOAS are better than in case of GOME2B (Table 5). However, the mean bias of GOME2A from the ground-based data is about half compared to the GOME2B bias over the RC site but slightly higher or similar over the SC and UC sites. Interestingly, the difference between GOME2A and GOME2B mean values (Table 4) is larger for the RC site and smaller for the SC and UC areas, which is not consistent with the pixel size influence. Especially in case of UC site a better detection of the higher $NO_2$ loadings in the city center by GOME2A should be expected, which, as can be seen also from the mean satellite $NO_2$ columns in Fig.5, is not the case.

### 3.3 Reconstruction of OMI observations

Hourly data of tropospheric $NO_2$ around the OMI overpass time derived from the CAMx simulations according to the method described in section 2.4 are presented for each campaign location separately in Fig. 9 (left panels), both as averages over the

close-to-actual OMI pixel and for the grid cell of 2 km × 2 km containing the location of the ground-based instrument. The AFs, i.e. ratios of the VCD at the single cell to the average, are also shown in Fig. 9 (right panels).

For the center of Thessaloniki (UC) the mean AF is 2.83±1.00, while at the other two locations the mean AF is much smaller. In the case of the RC site, the AF is on average close to unit (1.20±0.39). This indicates that the area within the OMI pixel corresponds mostly to lower $NO_2$ levels and similar to those at the MAX-DOAS location. Although for a different period, the majority of the AFs calculated for the SC site (middle row in Fig. 9) are slightly lower than 1, leading to an average of 0.75±0.39, which indicates higher average $NO_2$ concentrations over the OMI footprint than at the MAX-DOAS location. These results can be ascribed to the fact that the SC site is located about 13 km west of the center of Thessaloniki, hence, the OMI pixel area can be affected by heavy pollution levels observed in the city center. However, AFs larger than unit are observed occasionally, possibly related to local pollution sources or to $NO_2$ transportation by the prevailing winds from neighboring locations.

The wind direction and speed at the 3-hour interval around the OMI overpass time (10:00-13:00 UTC) during the campaign period are visualized in Fig. 10. The wind data set was obtained by a vector anemograph system installed at the Weather Station of Eptapyrgio-Thessaloniki (http://www.envdimosthes.gr), which is located 177m above sea level and ~800m to the north of the UC site. Due to its elevated location compared to the city area the wind data can be considered representative of the general wind field over the greater area, generally unaffected by local patterns. The wind during the period of the campaign is mostly of south-west direction with low to moderate speed, especially in spring months. Less frequent, stronger winds can be also observed, blowing mostly from northern and eastern directions which generally favor cleaning of the air over all sites.

The horizontal representativeness of the MAX-DOAS slant column measurements along the light path is found to be well within the 2 km × 2 km model cell (see also section 2.4). However, in order to investigate the effect on the AFs, we repeated the calculations for all hourly data of CAMx by taking the ratio of the VCD at the cell of each monitoring location to the VCD averaged over a typical OMI pixel at nadir (covering 7x13 CAMx cells) centered at this cell, as well as to the average VCD in two and three CAMx cells in the light path direction (Table 6). This way, the average AFs when using one cell are 2.20±0.75, 0.78±0.24, and 0.96±0.18 for UC, SC and RC site, respectively. These values are very similar to those calculated assuming a close-to-actual pixel size and position. The difference in the resulted mean factors when two cells are used, is about 1-2% in all cases except for the UC site during the OMI overpass time (difference of ~5%). When the averages of three cells are used in the calculations, the absolute differences are in general higher, e.g. up to about 5% in the SC site and 13% in the UC site. The much higher decrease in the UC site can be explained by the influence of the lower $NO_2$ concentration levels in some light path directions over the sea which is very close to the urban area (Fig. 6). It is worth mentioning that in the RC site the average ratio when all simulations are considered is around unity irrespective of the number of cells taken into account. In all other cases, except in the UC site when all simulations are considered, the differences in the resulting ratios using more than one cells are negative, which means that the average tropospheric $NO_2$ is lower when the light path is extended further away from the measurement location.

The comparison between ground-based and OMI observations over the monitoring locations before and after the adjustment of the satellite data is presented in Fig. 11 and Fig. 12. The MAX-DOAS and satellite data sets at the UC site are separated into the two different periods of monitoring at the RC and SC sites, so as the $NO_2$ data series for the different locations are directly comparable. Over the UC site the MAX-DOAS data compare much better with the adjusted OMI data. The average underestimation of OMI for the whole period of the campaign has decreased from $6.60\pm5.71 \times 10^{15}$ to $1.68\pm5.01 \times 10^{15}$ molecules $cm^{-2}$. There is substantial improvement in the comparison for the period November 2014 - January 2015, during which the $NO_2$ columns observed by the MAX-DOAS system are much higher compared to the satellite data. The underestimation of OMI is reduced from $10.67\pm4.53 \times 10^{15}$ to $4.99\pm3.30 \times 10^{15}$ molecules $cm^{-2}$. For the period January-May 2015 the difference is only slightly reduced from $-2.86\pm3.85 \times 10^{15}$ to $-1.35\pm4.41 \times 10^{15}$ molecules $cm^{-2}$, likely due to the close-to-background $NO_2$ concentration levels observed by both the MAX-DOAS and the satellite. Accordingly, the slope of the fitted least squares regression line has been improved from ~0.4 to ~0.9 and from ~0.2 to ~0.3 for the two periods, respectively. The OMI data over the other two sites have not been significantly affected by the adjustment, probably due to better spatial homogeneity of the $NO_2$ loading.

It should be noted here that uncertainties may arise from the vertical integration of the air quality simulations to derive the vertical column of $NO_2$ representative for the troposphere and the assumption that it is comparable to the VCD derived by the MAX-DOAS measurements. The two methods are based on different concepts, and for the MAX-DOAS it is not possible to define a top level for the tropospheric $NO_2$ column which could be used to integrate vertically the model simulations. The importance of this effect was investigated in a sensitivity analysis which showed that the uncertainty in the calculation of the VCD of $NO_2$ associated with the upper level used in the vertical integration of the model derived mixing ratio is less than 1% for different upper levels between 6 and 9 km.

In addition to the gradient smoothing effect, part of the underestimation of tropospheric $NO_2$ by OMI observations can be explained by the so-called aerosol shielding effect [Jin et al., 2016; and references therein], when an elevated layer of scattering aerosols lead to reduced satellite sensitivity in the lower troposphere [Leitão et al., 2010]. Especially, in the RC site, where the spatial gradients are weaker, some lower $NO_2$ values observed by OMI in winter can be explained only by the shielding effect. The aerosol shielding effect increases with AOD and SZA, thus, it is more efficient in winter than in summer, assuming the same AOD and aerosol profile [Ma et al., 2013]. Part of the aerosol shielding effect on OMI observations is avoided by the SZA filtering (SZA≤75º) applied in this study and, also, by the cloud correction algorithm and the used filtering with CLF (almost clear sky observations), since aerosols are partly detected as clouds and measurements with high aerosol optical depth might also be classified as cloudy measurements (Ma et al., 2013). In contrast to the shielding effect observed in space-borne measurements, the MAX-DOAS observations seem to overestimate the $NO_2$ VCD due to multiple scattering in elevated aerosol layers, but this overestimation is much less pronounced [e.g. Ma et al., 2013].

## 4 Summary and conclusions

An experimental campaign took place in the greater Thessaloniki area, Greece, within the period November 2014 – May 2015, with the aim to investigate the impact of different spatial sampling of ground-based and space borne measurements on tropospheric $NO_2$ column density. In this study, tropospheric $NO_2$ derived by the Phaethon ground-based MAX-DOAS systems at three different locations characterized by diverse pollution loadings were presented and compared with corresponding satellite products of OMI/Aura and GOME-2/MetOp-A and /MetOp-B. The main findings are:

- The agreement of MAX-DOAS measurements with satellite columns over the rural area is good for all three satellite instruments, although the variability is higher for GOME2B. The mean differences of OMI, GOME2A and GOME2B from the ground-based data are -1.63, -1.26 and -2.37 $\times 10^{15}$ molecules cm$^{-2}$, respectively.

- A strong negative bias in satellite $NO_2$ columns is found over the urban area (-6.60±5.71, 10.51±8.32 $\times 10^{15}$ and 10.21±8.87 $\times 10^{15}$ molecules cm$^{-2}$ for OMI, GOME2A and GOME2B, respectively). This is attributed to the great inhomogeneity of $NO_2$ concentrations over the area covered by the city of Thessaloniki. A significant portion of the area surrounding the city center and included in a typical sub-satellite pixel corresponds mostly to rural atmospheric conditions. Therefore the average tropospheric $NO_2$ is significantly reduced compared to Phaethon data that probe air masses mainly over the center of the city.

- Our findings from the comparison between MAX-DOAS and OMI data over the UC and SC (r=0.77 for UC and r=0.54 for SC) are similar to the results of previous validation studies [e.g. Kramer et al., 2008; Ialongo et al., 2016]. However, we have calculated smaller correlation coefficients between the MAX-DOAS and GOME-2 data (0.28 for GOME2A and 0.19 for GOME2B at the UC) compared to other studies [e.g. Valks et al., 2011] due to more complex air pollution patterns over Thessaloniki.

- The agreement of Phaethon data with OMI retrievals is better compared to GOME-2 sensors, due mainly to its smaller footprint. For GOME2A and GOME2B the city center is a very small portion compared to the entire area covered by the satellite pixel, thus the contribution of the increased $NO_2$ loadings to the average column density is very weak. The differences between the two satellite retrieval algorithms as well as the different overpass times also play an important role. Differences observed between GOME2A and GOME2B tropospheric $NO_2$ can be explained by their different pixel sizes and instrumental degradation.

- The $NO_2$ time series over the UC site was also investigated separately for the different periods of measurements at the RC and SC sites, i.e. from 1 November 2014 through 31 January 2015 and from 20 January through May 2015 respectively. Interestingly, the satellite and ground-based data show better agreement during the latter period due to lower $NO_2$ concentrations observed in the urban area in spring. The mean difference of OMI from the MAX-DOAS over the UC was found to be -10.67 $\times 10^{15}$ and -2.86 $\times 10^{15}$ for the two different periods, respectively.

- The application of adjustment factors derived from $NO_2$ simulations with an air quality modeling tool on OMI data reduced the effect of spatial differences between OMI and Phaethon observations. The improvement in the

comparisons is more evident for the urban site, where the mean difference between OMI and Phaethon for the whole campaign period was reduced from $6.60\pm5.71 \times 10^{15}$ to $1.68\pm5.01 \times 10^{15}$ molecules cm$^{-2}$. The effect on the rural and suburban locations is negligible, due to the reduced spatial inhomogeneity of the air-pollution.

In the present study, the data set of the MAX-DOAS instruments that was used for the validation of the satellite observations was rather short due to technical issues that arose during the campaign, and has not allowed more thorough investigations. However, the applied method for the adjustment of OMI data has significantly reduced the satellite underestimation and improved the comparisons with the MAX-DOAS over the urban area, suggesting that the application of such procedures will improve the satellite validation results over areas with complex spatial distribution of air pollutants.

**Acknowledgements:** This study was conducted in the framework of the QA4ECV (Quality Assurance for Essential Climate
Variables) project, that was financed under theme 9 (Space) of the European Union Framework Program 7. The three Phaethon systems have been developed in the framework of the Operational Program "Education and Lifelong Learning" of the National Strategic Reference Framework (NSRF) Research Funding Program: ARISTEIA I - 608, AVANTI (Optimization and expansion of ground infrastructure for the validation of satellite-derived column densities of atmospheric species) project, that was co-financed by the European Union (European Social Fund-ESF) and Greek national funds. The authors would like to
thank the Earth Observation Center, Remote Sensing Technology Institute, German Remote Sensing Data Center, DLR, for the dissemination of the GOME2/MetopA data, the NASA Earth Science Division for funding the OMI NO$_2$ development and the Environment Division of the Municipality of Thessaloniki for providing the wind speed and direction data set. C-IFS-TM5 global model system data were provided in the framework of the EU project Monitoring Atmospheric Composition and Climate III (MACC-III) (Grant agreement no: 633080). Finally, we would like to thank TNO for the European scale
anthropogenic emission data provided in the framework of the EU project Monitoring Atmospheric Composition and Climate (MACC) (Grant agreement no.: 218793).

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

**Table 1. Information on campaign locations and measurements performed there.**

|  | LAP-AUTH (UC) | ATEITH (SC) | Epanomi (RC) |
|---|---|---|---|
| Air quality conditions | Urban | Sub-urban/Industrial | Rural |
| Latitude | 40.63N | 40.65N | 40.37N |
| Longitude | 22.96E | 22.81E | 22.98E |
| System | Phaethon #1 | Phaethon #3 | Phaethon #2 |
| Period of operation | 1/11/2014 - 11/5/2015 | 20/1/2015 - 11/5/2015 | 1/11/2014 - 31/1/2015 |
| Number of $NO_2$ trop. VCD measurements | 7930 | 16716 | 5738 |

**Table 2. Tropospheric $NO_2$ statistics from the comparison of Phaethon #2 and #3 systems to Phaethon #1 during the period 11 - 19 October 2014. The hourly averaged vertical columns at both 15° and 30° elevation viewing angles are compared.**

|  | Phaethon #2 | Phaethon #3 |
|---|---|---|
| Number of observations | 75 | 75 |
| Correlation coefficient (r) | 0.95 | 0.92 |
| Slope of the linear fit | 1.03 | 1.07 |
| Mean bias [x$10^{15}$ molec. cm$^{-2}$] | 0.58 | 0.73 |
| Standard deviation (1σ) [x$10^{15}$ molec. cm$^{-2}$] | 2.51 | 3.12 |

**Table 3: Main features of the satellite algorithms.**

|  | DLR GDP 4.7 (GOME2) | NASA V3 (OMI) |
|---|---|---|
| SCD Retrieval Methodology | DOAS within 425-450nm [Platt, 1994; Platt and Stutz, 2008] | DOAS within 402-465nm [Marchenko et al., 2015] |
| Stratospheric component | Spatial filtering [Wenig et al., 2004] / masking of polluted areas [Loyola et al., 2007; Valks et al., 2011] | In regions of tropospheric pollution, stratospheric column is inferred using a local analysis of the stratospheric field [Celarier et al.,2008; Bucsela et al., 2013] |
| Cloud algorithm | OCRA/ROCINN version 2.0 [Loyola et al., 2007] | Updated $O_2$–$O_2$ [Acarreta et al., 2004] |
| AMF calculation | LUT generated using the Linearized Discrete Ordinate Radiative Transfer (LIDORT) Model [Palmer et al., 2001; Spurr et al., 2001; 2008] | LIDORT Model [Palmer et al., 2001; Spurr et al., 2001; 2008] |
| $NO_2$ a-priori profile | Monthly mean profiles derived from MOZART -2 CTM [Horowitz et al., 2003] | Monthly mean profiles derived from Global Modeling Initiative CTM [Douglass et al., 2004] |
| Main algorithm reference | Valks et al. [2011] or Hassinen et al. [2015] | Celarier et al. [2008], Bucsela et al. [2013; 2016] |

**Table 4. Average and standard deviation of space-borne and ground-based tropospheric NO₂ observations over each monitoring site. Phaethon mean values have been calculated from measurements at 15° and 30° elevation viewing angles (see also section 2.2) within ±30 min around the satellite overpass time. The NO₂ measurements at the UC site are separated into two periods: from November 2014 through end of January 2015 and from mid-January through mid-May 2015.**

| NO₂ trop. VCD mean (±1σ) [x10$^{15}$ molec. cm$^{-2}$] from | RC site | SC site | UC site | |
|---|---|---|---|---|
| | | | Nov 2014 – Jan 2015 | Jan - May 2015 |
| OMI | 3.19 (±1.39) | 2.18 (±1.26) | 3.92 (±2.84) | 2.15 (±1.14) |
| Phaethon (OMI overpass time) | 4.82 (±2.69) | 2.35 (±1.05) | 14.59 (±6.33) | 5.01 (±4.60) |
| GOME2A | 3.04 (±1.58) | 2.17 (±1.25) | 3.39 (±1.58) | 2.49 (±1.39) |
| Phaethon (GOME2A overpass time) | 4.30 (±1.63) | 4.77 (±3.03) | 16.43 (±8.56) | 10.08 (±8.53) |
| GOME2B | 3.33 (±2.50) | 2.35 (±1.19) | 3.25 (±2.13) | 2.56 (±1.22) |
| Phaethon (GOME2B overpass time) | 5.70 (±2.23) | 4.66 (±2.84) | 15.53 (±9.64) | 10.18 (±7.49) |

**Table 5. Statistics from the comparison of space-borne and ground-based NO₂ tropospheric VCDs for each monitoring site. The same MAX-DOAS data were used as in Table 4.**

| | Compared to Phaethon NO$_2$ trop. VCD | RC site | SC site | UC site | |
|---|---|---|---|---|---|
| | | | | Nov 2014 – Jan 2015 | Jan - May 2015 |
| OMI | Number of collocations | 11 | 19 | 11 | 12 |
| | Correlation coefficient (r) | 0.37 | 0.54 | 0.77 | 0.73 |
| | Slope | 0.19 | 0.65 | 0.35 | 0.18 |
| | Intercept [x10$^{15}$ molec. cm$^{-2}$] | 2.28 | 0.66 | -1.12 | 1.24 |
| | Mean bias [x10$^{15}$ molec. cm$^{-2}$] | -1.63 | -0.17 | -10.67 | -2.86 |
| | Standard deviation (1σ) [x10$^{15}$ molec. cm$^{-2}$] | 2.54 | 1.12 | 4.53 | 3.85 |
| GOME2A | Number of collocations | 22 | 34 | 24 | 17 |
| | Correlation coefficient (r) | 0.44 | 0.02 | 0.01 | 0.26 |
| | Slope | 0.42 | 0.01 | 0.003 | 0.04 |
| | Intercept [x10$^{15}$ molec. cm$^{-2}$] | 1.22 | 2.12 | 3.35 | 2.06 |
| | Mean bias [x10$^{15}$ molec/cm$^{2}$] | -1.26 | -2.60 | -13.04 | -7.59 |
| | Standard deviation (1σ) [x10$^{15}$ molec. cm$^{-2}$] | 1.70 | 3.25 | 8.68 | 8.27 |
| GOME2B | Number of collocations | 27 | 52 | 29 | 27 |
| | Correlation coefficient (r) | -0.10 | -0.07 | 0.18 | -0.06 |
| | Slope | -0.11 | -0.03 | 0.04 | -0.01 |
| | Intercept [x10$^{15}$ molec. cm$^{-2}$] | 3.97 | 2.48 | 2.63 | 2.66 |
| | Mean bias [x10$^{15}$ molec/cm$^{2}$] | -2.37 | -2.31 | -12.28 | -7.62 |
| | Standard deviation (1σ) [x10$^{15}$ molec. cm$^{-2}$] | 3.51 | 3.16 | 9.49 | 7.65 |

**Table 6. Average and standard deviation of the CAMx simulated NO$_2$ tropospheric VCDs at the 2 km × 2 km cell containing each monitoring site and for the OMI nadir pixel (14 km x 26 km) centered at each site for the entire period of the campaign, separately for all simulations and for simulations near the OMI overpass time. The corresponding mean adjustment factors (AF) for two and three model cells in the light path direction are also presented.**

| Mean (±1σ) NO$_2$ trop. VCDs [x10$^{15}$ molec. cm$^{-2}$] and adjustment factors from | | RC site | SC site | UC site |
|---|---|---|---|---|
| All simulations | 2 km x 2 km | 3.70 (±2.74) | 5.02 (±3.73) | 11.58 (±8.43) |
| | 14 km x 26 km | 3.59 (±2.37) | 5.73 (±3.22) | 5.87 (±3.36) |
| | AF (single cell) | 1.00 (±0.16) | 0.85 (±0.25) | 1.91 (±0.75) |
| | AF (2 cells) | 1.00 (±0.16) | 0.83 (±0.23) | 1.93 (±0.66) |
| | AF (3 cells) | 1.00 (±0.15) | 0.81 (±0.21) | 1.83 ( ±0.60) |
| Simulations near OMI overpass time | 2 km x 2 km | 2.85 (±1.97) | 3.62 (±2.64) | 12.43 (±8.66) |
| | 14 km x 26 km | 2.85 (±1.55) | 4.50 (±2.36) | 5.46 (±3.12) |
| | AF (single cell) | 0.96 (±0.18) | 0.78 (±0.24) | 2.20 (±0.75) |
| | AF (2 cells) | 0.95 (±0.18) | 0.77 (±0.23) | 2.09 (±0.66) |
| | AF (3 cells) | 0.94 (±0.17) | 0.76 (±0.22) | 1.92 (±0.62) |

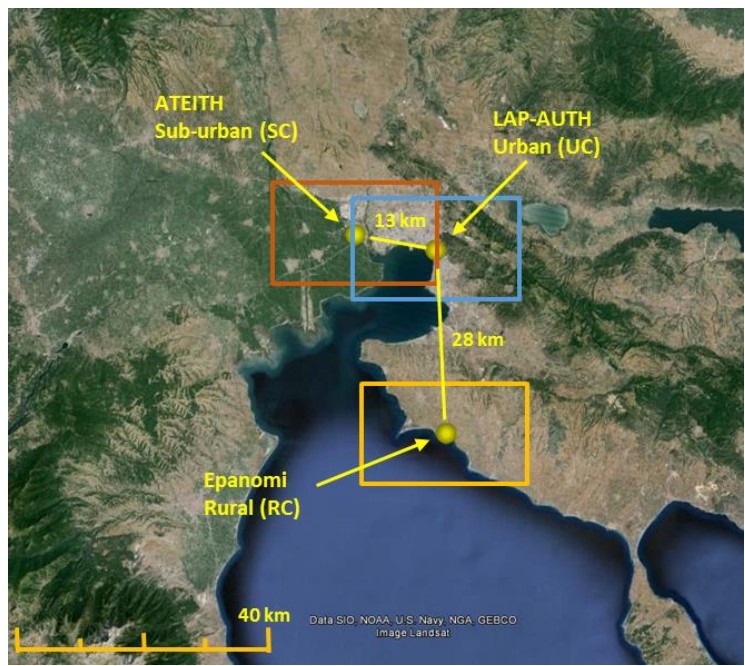

**Figure 1: Map of the greater Thessaloniki area with the three sites of measurements: UC (blue), SC (red) and RC (yellow). The rectangular outlines represent the area covered by the nadir pixel size of OMI centered at each location (Courtesy of Google Earth NASA Images).**

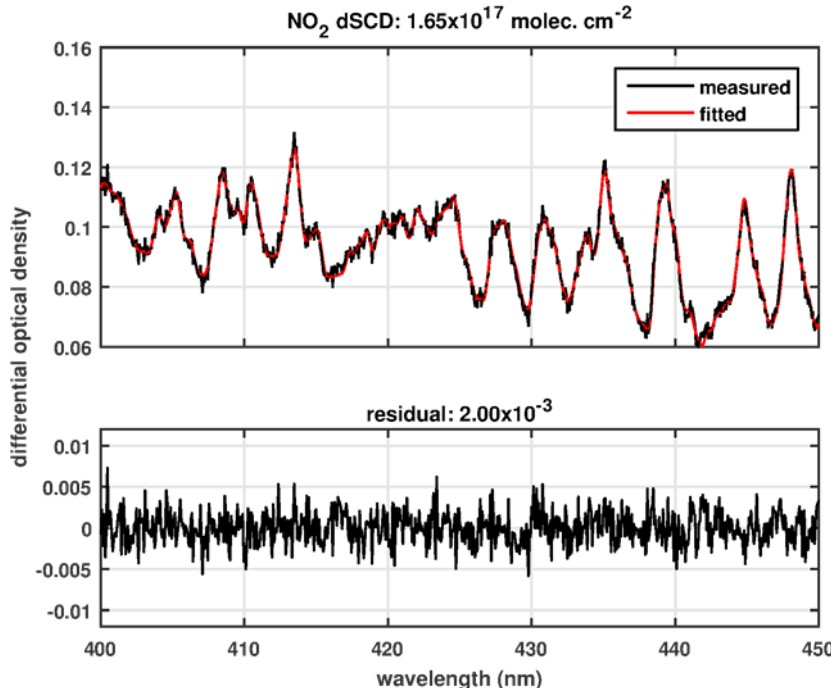

**Figure 2: Example of NO₂ fitting results obtained at the UC site on 6 November 2014, around 10:30 UTC, at an elevation angle of 5º and a SZA ~57º. The upper panel shows the measured (black) and the fitted (red) NO₂, and the lower panel shows the residual of the DOAS fit.**

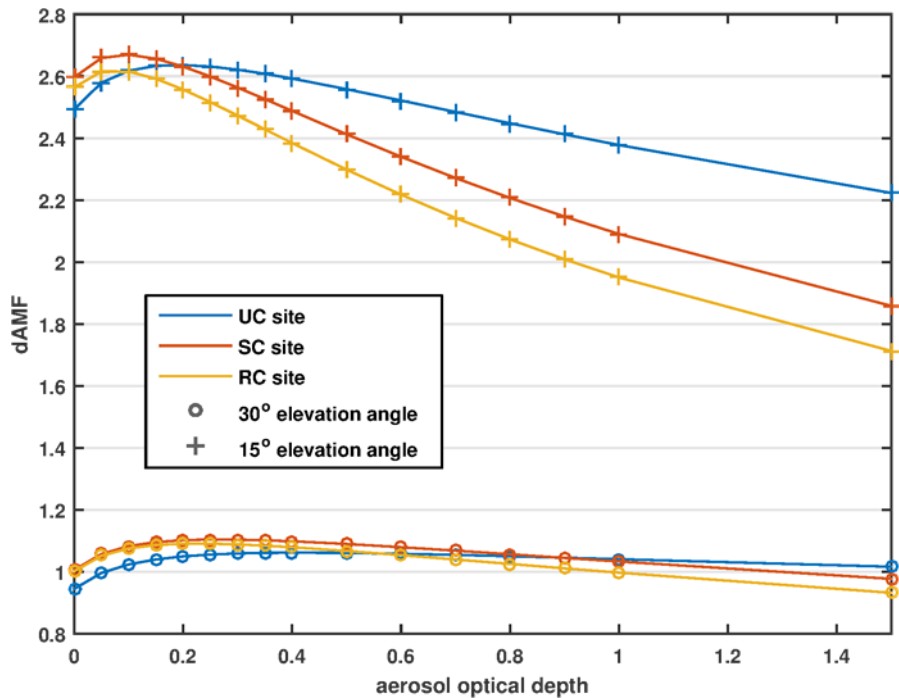

**Figure 3: Example of AMFs calculated with libRadtran at SZA 40° and azimuth angle 80° relative to the sun for each campaign site. The AMFs at 15° (crosses) and 30° (circles) elevation angles are plotted versus the AOD.**

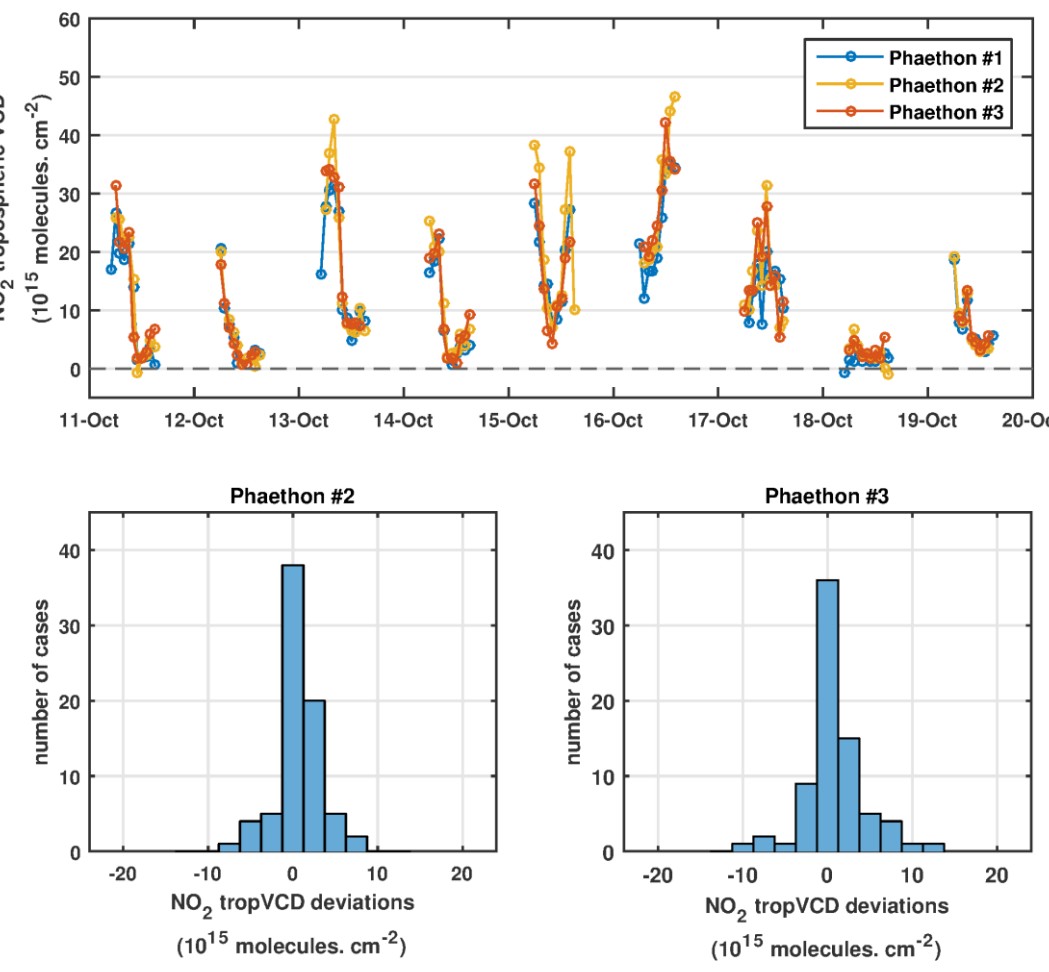

**Figure 4: The Phaethon systems were operating in parallel in the University Campus from 11 to 19 October 2014. Upper panel: Hourly mean NO₂ tropospheric VCD time series of the three systems retrieved for 15° and 30° elevation angles. Lower panels: Histograms of the NO₂ tropospheric VCD differences of Phaethon #2 and #3 relative to Phaethon #1.**

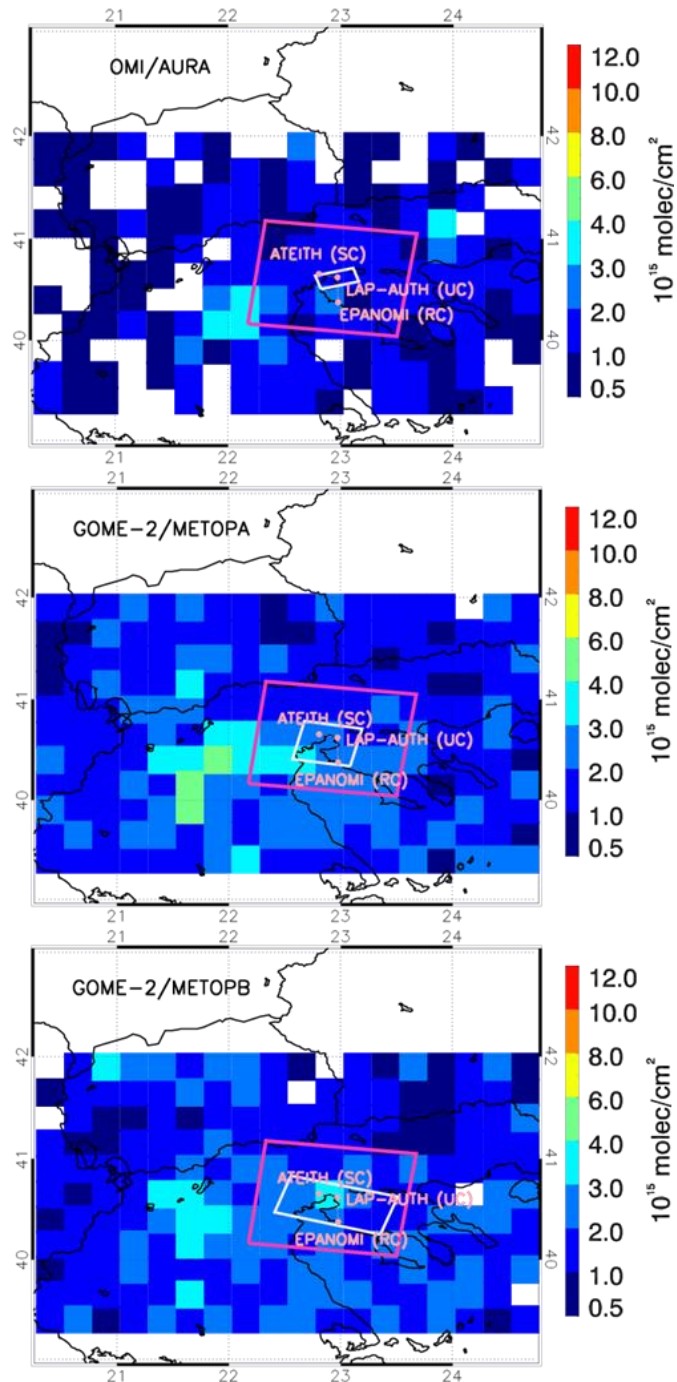

**Figure 5: Maps of the tropospheric NO₂ spatial distribution averaged over the campaign period from OMI (upper panel), GOME2A (middle panel) and GOME2B (lower panel) observations. The white outlined areas represent the GOME-2 and OMI pixel sizes for the 11th and 12th of November 2014, respectively. The pink outlined areas represent the CAMx domain used in Figure 6. Blank cells indicate lack of sufficiently good quality data.**

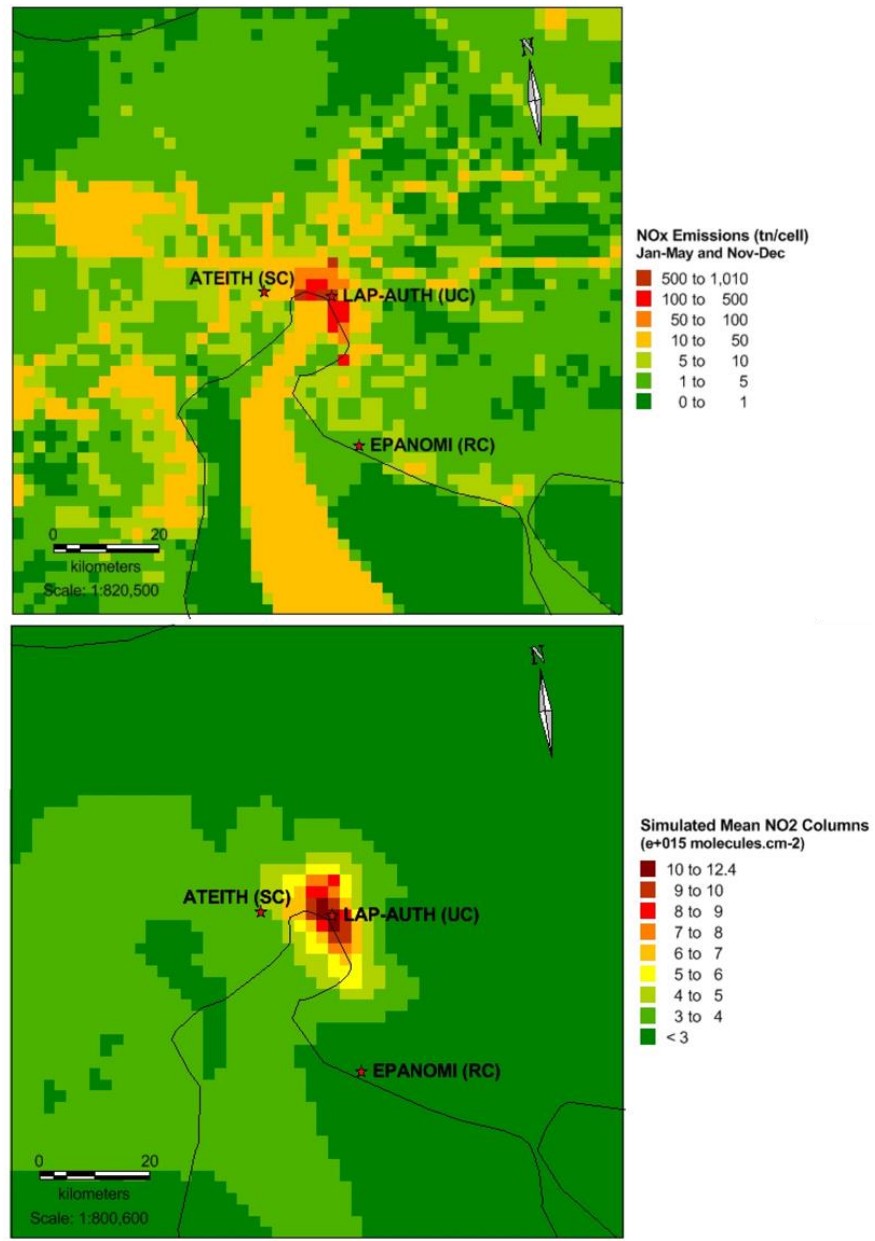

**Figure 6: The upper panel shows the total NOx emissions of the period November to May for the domain of Thessaloniki. The lower panel shows the simulated NO₂ VCD averaged over the period of the campaign for the same domain.**

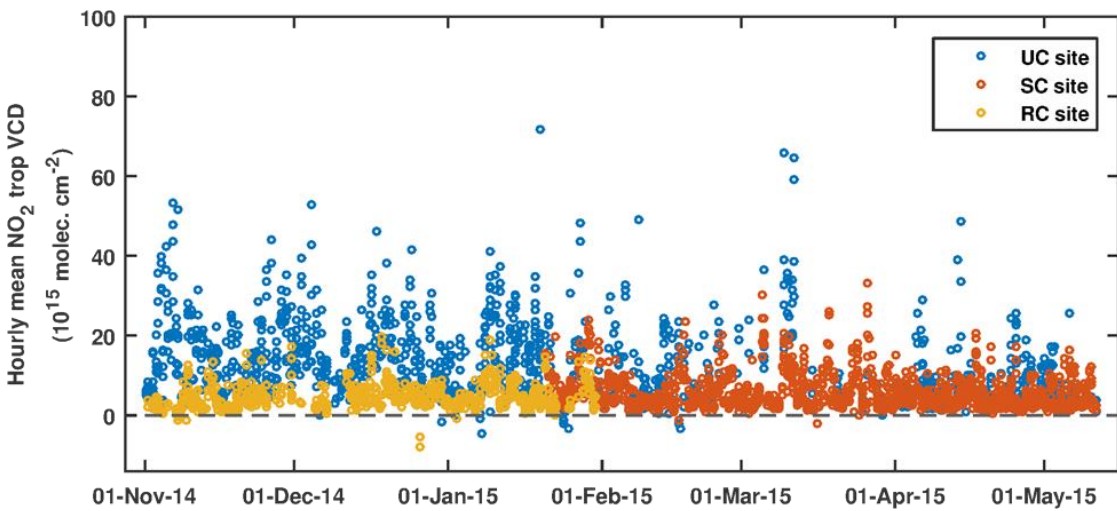

**Figure 7: Time series of hourly mean tropospheric NO$_2$ VCD measurements performed at 15° and 30° elevation angles by Phaethon at the three campaign sites.**

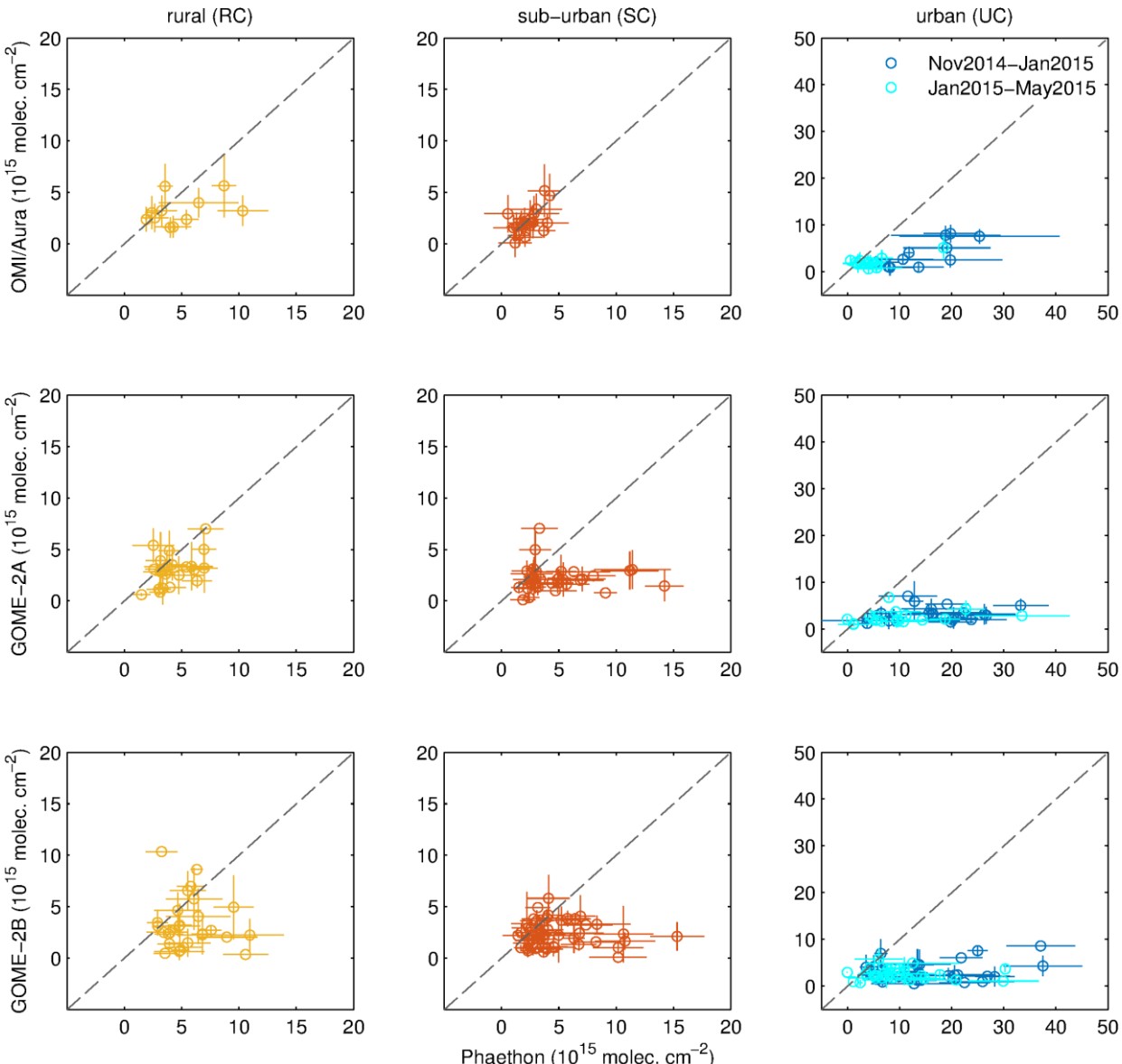

**Figure 8: Scatter plots of tropospheric NO₂ VCD derived from the ground and space for each campaign site (RC left column, SC middle column, and UC right column) and satellite sensor (OMI top row, GOME2A middle row and GOME2B bottom row). The NO₂ measurements at the UC site are separated into two periods: from November 2014 through end of January 2015 (blue circles) and from mid-January through mid-May 2015 (light blue circles). Error bars are the standard deviation of all data points entering the mean for the ground-based data and the estimated total error for the satellite overpass data. Note the different scale for the UC site. Statistics of the comparisons can be found in Tables 4 and 5.**

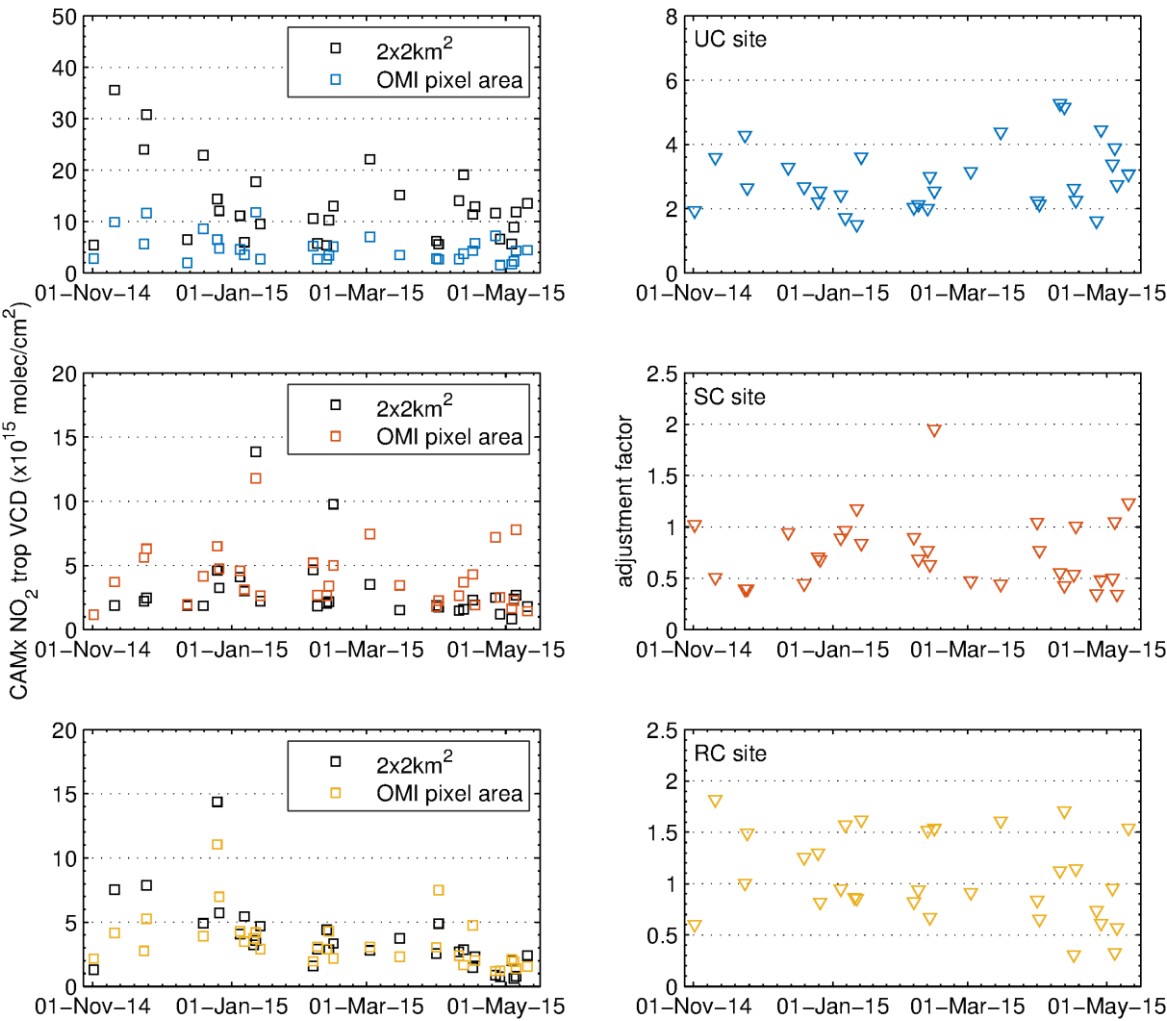

**Figure 9:** Hourly NO₂ tropospheric VCDs (left column) and the corresponding adjustment factors (right column) calculated by means of CAMx air quality simulations for the UC site (top row), the SC site (middle row) and the RC site (bottom row). Left panels: Only the NO₂ columns from CAMx corresponding to OMI overpass data are presented here. The black squares correspond to data for the CAMx cells which include the monitoring sites and the colored squares to data averaged over the close-to-actual OMI pixel area.

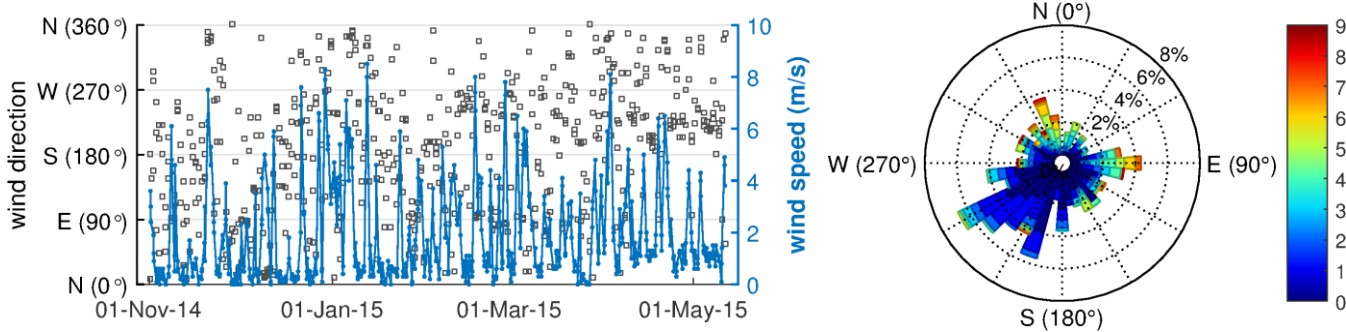

**Figure 10: Time series of wind direction and speed hourly data at the three-hour interval (10:00-13:00 UT) around the OMI overpass time during the campaign period (left panel) and the corresponding wind rose diagram (right panel).**

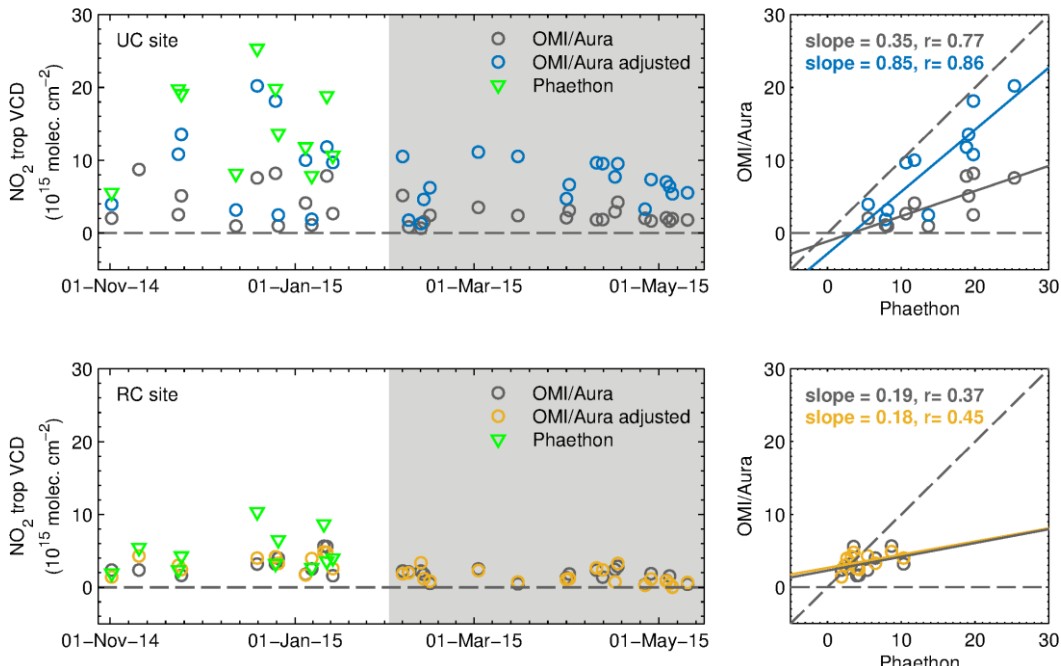

**Figure 11: Time series of tropospheric NO₂ over the UC and RC sites derived from Phaethon and OMI before and after the adjustment of OMI data (left panels). From the measurements at the UC site only those performed for the period from 1 November 2014 through 31 January 2015 are shown. The corresponding scatter plots are shown in the right panels.**

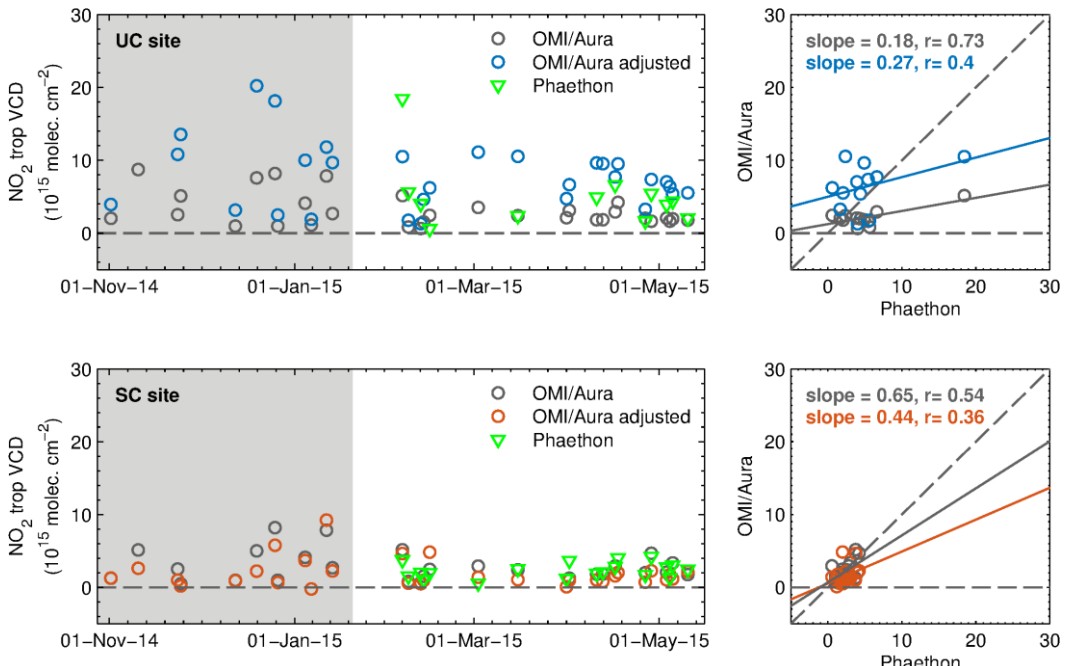

**Figure 12: Same as Figure 11 but for the UC ad SC sites and the period 20 January to 11 May 2015.**