# Peer review of "Comparisons of ground-based tropospheric NO2 MAX-DOAS measurements to satellite observations with the aid of an air quality model over Thessaloniki area, Greece"

_Atmospheric Chemistry and Physics, 2016_

## Referee Comment (RC1) · Anonymous Referee #1 · 4 Oct 2016

**Interactive comments on "Comparisons of ground-based tropospheric NO2 MAX-DOAS measurements to satellite observations with the aid of an air quality model over Thessaloniki area, Greece" by T. Drosoglou et al.**

**General comments:**

This paper presents a comparison of satellite observations to ground based tropospheric NO2 MAX-DOAS measurements performed in 3 different types of conditions (urban, suburban and rural). A high resolution air quality model is used to try to assess the difference in spatial resolutions between the OMI satellite data and the MAXDOAS.

The paper is well written and the comparison is of interest of the community, however some of the data are not exploited/commented enough and I recommend the publication after the suggested revisions.

One of the major pity is that the air quality model is used only for the OMI results and not for the GOME-2 data. Moreover, the air quality model is used with the shortcoming of a typical OMI nadir pixels centered around each location (p.10, line 7 to 13) instead of the actual pixel size and location. If additional simulations are unavailable to improve the study, the variability of the NO2 content in space should be estimated in some way (maybe from the NOx emissions) and discussion of the spatial and temporal variability should be enhanced. For GOME-2 comparisons, the fact that the pixels would cover the whole domain of Fig. 1 (p. 7, line 33) can be used to test how the average variability seen by 3 typical stations would compare to the averaged variability seen from satellite. Moreover, the differences between GOME-2A and GOME-2B comparisons results are never mentioned, while Metop-A is in a reduced swath mode since 7/2013 with a pixel half the size of the nominal mode of Metop-B. This should be mentioned and commented (figure 7, table 3 and 4). The OMI and GOME-2 tropospheric NO2 algorithms are only mentioned in Section 2.3, without any detailed description, and conclusions are thrown as if they were consistent among them, while we know that large differences in tropospheric NO2 exist between different product (applied on a same instrument) due to different algorithm hypothesis. The 2 algorithms should be described in Section 2.3 (maybe with a table of main differences) and impact of the algorithm hypothesis should be mentioned/quantified. Comparison e.g. of the model NO2 profiles at the 3 stations type wrt to the profiles used as a-priori in the satellites retrievals would be interesting.

**Specific comments and technical corrections**

Page 1, line 9: consider changing "the main" into "one of the"

Page 1, line 13 (and in the conclusions): why using DOAS/MAX-DOAS while in the rest of the document only "MAX-DOAS" is used?

Page 1, line 18: consider reformulating "mainly due to the higher spatial resolution of OMI" to also mention the different products.

Page 2, line 22: consider changing "is limited… respectively" to "are respectively limited by the …"

Page 3, line 10: "are applied to satellite data …" → Only applied to OMI data!

Page 4, line 18: inconsistency between "a fixed azimuth angle of 255º" and "at azimuth angles of 80° relative to the solar azimuth". Reformulate.

Page 4, line 21: add reference for "only at the elevation angles of 15º and 30º in order to avoid uncertainties introduced due to aerosol loadings at lower elevation angles" and explain what is done with the VCD retrieved at the 2 elevation angles. Is the average value used? Or is a filtering related to the difference of the 2 VCD values used (as in Brinksma et al., 2008)?

Page 5, line 7: illustrate the shape of the used "mean vertical profiles from the air quality modelling tool consisting of the photochemical grid model CAMx and the mesoscale weather prediction system WRF" (ideally also showing how they differ from the 2 satellite NO2 algorithms a-priori profiles)

Page 5, line 13: the 0.07 and 0.01 difference in AOD between the sites should be referred to the mean value. (0.07 wrt to 0.1 is large while it is small if compared to 1).
Why scaling the LIDAR profile to AOD values (line 14), if the "AOD loading and AOD profile between the three locations is very small" (line 12-13)? How much the choice of this aerosols a-priori profile affects the MAXDOAS NO2 VCD? Give an estimation of the MAXDOAS error.
Maybe add a subplot on figure 6 with the time-series of the Thessaloniki CIMEL AOD to have an idea of the variability?

Page 5, line 18: it's Fig 4 and not 3. Line 21 it's Fig 2 and not 4.

Page 5, line 24-25: I would put this sentence at the end of the section.

Section 2.3: add a description of the 2 NO2 algorithms maybe with an overview table) and references to papers (at least Valks et al., 2011 and Bucsela et al., 2013) and some other validation results (here on when discussing the results).

Page 6, line 17: "a particular direction and path" – how long is the representativeness area of the MAXDOAS? Give references (at least irie et al., 2011) and estimation ranges. In page 7, line 32, an area of 2x2km² is considered representative of the MAXDOAS data, but when using high elevation angles as 15° and 30°, and considering no aerosols, longer distances are obtained. How much the choice of using only one model cell of 2x2km² is affecting the adjustment factor?

Page 6, line 19: "the satellite data are adjusted" → only OMI data are adjusted.

Page 6, line 25 to page 7 line 24: a lot of details are given for the model data used here, which is too much compared to the lack of description of the satellite data in Sect 2.3. The goal of the papier is focusing on the satellite data!

Page 7, line 25: again, only OMI data are adjusted.

Page 7, line 32: see comment above about the impact of using only one model cell of 2x2km² is affecting the adjustment factor.

Page 7, line 33: "the method was applied only to OMI data because for GOME-2 the sub-satellite pixel is very large, covering typically the entire domain of Fig. 1" – see general comment: how the GOME-2 data would compare to an average of the 3 stations?

Page 8, line 14 and line 19: again, what columns are used in this study (in the figures, tables, etcc)? VCD at 15° or at 30°? Or something else?

Page 8, line 26: the "upper limit for the distance": does this mean that only the closest pixel within this distance is used for the satellite data? Or an average of the pixels values within this distance is used? How much choosing 50km for both instruments would affect the comparison?

Page 8, line 29: typo: "table 3 and 4"

Page 9, line 6: "can be attributed mainly to its smaller pixel size" and smaller distance criterium?
"to its higher sensitivity in the boundary layer compared to GOME-2": give a reference.
The fact that the 2 algorithms are different should also be mentioned and commented.

Page 9, line 11: "The use of actual satellite geometry for each day is complex and would require a much larger domain for the air-quality simulations, more than double the currently used (120x120 km2) in order to include all possible pixel sizes and positions for each location. However, such simulations were not available for this study." → This is really a pity. The study would be much more realist…

Page 9, line 19: again, it would be nice to have a figure with the NO2 model profile at the 3 sites.

Section 4: no discussion to previous validation results is included, neither in Section 3 or 4. This should be added. Discussion of the difference in GOME2A and GOME2B results should also be added. Again importance of the OMI and GOME2 algorithmic differences should be mentioned.

Page 18, table 2: mention that the statistics are for tropospheric NO2. From which angle (12 or 30°)?

Page 18, table 3: discuss the larger differences in GOME2A wrt GOME2B mean values in RC sites and smaller in SC and UC (not consistent with the pixel size influence, as GOME2A is twice as small than GOME2B). A figure with location of the satellite pixels position/size could be useful.

Page 19, table 4: add slope and intercept values to this table. Again, discuss GOME2A wrt GOME2B differences.

Page 19, table 5: considering adding a line with values over 2 or more cells in the MAXDOAS pointing direction (related to comment on the MAXDOAS representativeness).

Page 20, figure 1: add a ruler to estimate distances.

Page 23, caption of figure 4: "and azimuth angle 100º relative to the sun for " → in the text (page 4, line 19) it's 80° !

Page 24, figure 5: it would be good to add the locations of the 3 stations also on the 1rst panel, and add a ruler to estimate distances.

Page 25, figure 6: why not including the period where the 3 instruments measured at the same site on the figure?

Page 26, figure 7: use the same axis limit for the urban scatter plot for the 3 instruments.

References:

Irie, H., Takashima, H., Kanaya, Y., Boersma, K. F., Gast, L., Wittrock, F., Brunner, D., Zhou, Y. and Van Roozendael, M.: Eight-component retrievals from ground-based MAX-DOAS observations, Atmos. Meas. Tech., 4(1), 1027–1044, doi:10.5194/amtd-4-639-2011, 2011.

Bucsela, E. J., Krotkov, N. A., Celarier, E. A., Lamsal, L. N., Swartz, W. H., Bhartia, P. K., Boersma, K. F., Veefkind, J. P., Gleason, J. F., and Pickering, K. E.: A new stratospheric and tropospheric NO2 retrieval algorithm for nadir-viewing satellite instruments: applications to OMI, Atmos. Meas. Tech., 6, 2607–2626.

Valks, P., Pinardi, G., Richter, A., Lambert, J.-C., Hao, N., Loyola, D., Van Roozendael, M. and Emmadi, S.: Operational total and tropospheric NO2 column retrieval for GOME-2, Atmos. Meas. Tech., 4, 1491–1514, doi:doi:10.5194/amt-4-1491-2011, 2011.

---

## Referee Comment (RC2) · Anonymous Referee #2 · 23 Oct 2016

Interactive comments on "Comparisons of ground-based tropospheric NO2 MAX-DOAS measurements to satellite observations with the aid of an air quality model over Thessaloniki area, Greece" by Theano Drosoglou.Interactive comments on "Comparisons of ground-based tropospheric NO2 MAX-DOAS measurements to satellite observations with the aid of an air quality model over Thessaloniki area, Greece" by Theano Drosoglou.

In this manuscript, the authors report on a comparison of ground based tropospheric NO2 MAXDOAS measurements to satellite observations with the benefit of an air qual-

ity model. The main point is the spatial resolutions between different instruments, for this purpose, different satellite data and model data were used.

The comparison of the MAX-DOAS data with satellite data is marginal fair. The main point of the comparison and rescaling is really fine but a better illustration of the context would be desirable. I recommend the publication after revisions as suggested here and the last review, please follow the general comments.

Comments: The map of satellite can show the variability of NOx and corresponding the emissions, this means, if the applied OMI data shows really the emitted from this area or not. The comparison of the NO2 tropospheric from different sites (UC, RC and SC) is less relevant because the data are from different time periods, for such comparison, you need the same time period. The adjustment factors are more depend on the model data than OMI data; this rescaling can be used but it is somehow banal. Without any adjustment factors, the results can be seen from fig. 4, what I expect for OMI reconstructed. Generally, you can compare every ground based with every satellite with different time periods but in this case, you can not compare the results together.

P4, l16: direct sun light, we have only scattered sun light, but if we look directly in the sun (still scattered), the measured spectra have structures, which should be removed from the analysis. P5,l5: albedo: 0.1. The three different areas (UC, SC and RC) have definitely different albedos. Fig. 5 down: Some pixels are missing after averaging! Fig. 7: It is not clear to me, if you used all data from different campaigns with different time periods or not, if yes, they are not comparable in such form . You need a significant criteria, namely the same time period. Fig. 9: You can not compare the slopes from different time periods specially with seasonal difference!

In this manuscript, the authors report on a comparison of ground based tropospheric NO2 MAXDOAS measurements to satellite observations with the benefit of an air quality model. The main point is the spatial resolutions between different instruments, for
this purpose, different satellite data and model data were used.

The comparison of the MAX-DOAS data with satellite data is marginal fair. The main point of the comparison and rescaling is really fine but a better illustration of the context would be desirable. I recommend the publication after revisions as suggested here and the last review, please follow the general comments.

Comments: The map of satellite can show the variability of NOx and corresponding the emissions, this means, if the applied OMI data shows really the emitted from this area or not. The comparison of the NO2 tropospheric from different sites (UC, RC and SC) is less relevant because the data are from different time periods, for such comparison, you need the same time period. The adjustment factors are more depend on the model data than OMI data; this rescaling can be used but it is somehow banal. Without any adjustment factors, the results can be seen from fig. 4, what I expect for OMI reconstructed. Generally, you can compare every ground based with every satellite with different time periods but in this case, you can not compare the results together.

P4, l16: direct sun light, we have only scattered sun light, but if we look directly in the sun (still scattered), the measured spectra have structures, which should be removed from the analysis. P5,l5: albedo: 0.1. The three different areas (UC, SC and RC) have definitely different albedos. Fig. 5 down: Some pixels are missing after averaging! Fig. 7: It is not clear to me, if you used all data from different campaigns with different time periods or not, if yes, they are not comparable in such form . You need a significant criteria, namely the same time period. Fig. 9: You can not compare the slopes from different time periods specially with seasonal difference!

Please also note the supplement to this comment:
http://www.atmos-chem-phys-discuss.net/acp-2016-611/acp-2016-611-RC2-supplement.pdf

---

## Author Comment (AC1) · 23 Jan 2017

**Response to anonymous referee #1**

We would like to acknowledge the referee for their helpful and thorough review. We believe that their comments improved the quality of this work.

Some of the measurements and statistics presented in the study have changed as a result of some of the reviewers' suggestions. Moreover, an updated OMI overpass data set has been used, generated by the Aura Validation Data Center (AVDC) (http://avdc.gsfc.nasa.gov) in November 28th 2016.

Our responses (in blue) follow the reviewer's comments (in black italics).

**General comments:**

*One of the major pity is that the air quality model is used only for the OMI results and not for the GOME-2 data. Moreover, the air quality model is used with the shortcoming of a typical OMI nadir pixels centered around each location (p.10, line 7 to 13) instead of the actual pixel size and location. If additional simulations are unavailable to improve the study, the variability of the $NO_2$ content in space should be estimated in some way (maybe from the NOx emissions) and discussion of the spatial and temporal variability should be enhanced. For GOME-2 comparisons, the fact that the pixels would cover the whole domain of Fig. 1 (p. 7, line 33) can be used to test how the average variability seen by 3 typical stations would compare to the averaged variability seen from satellite. Moreover, the differences between GOME-2A and GOME-2B comparisons results are never mentioned, while Metop-A is in a reduced swath mode since 7/2013 with a pixel half the size of the nominal mode of Metop-B. This should be mentioned and commented (figure 7, table 3 and 4).*

The NOx emissions pattern during the campaign period is discussed in section 2.4. GOME-2 observations have been compared to the average of the three stations and the results indicate that this average seems to be affected mostly by the much higher $NO_2$ concentrations of the urban site. Information on the reduced swath width of GOME2A has been included in section 2.3 and discussed in sections 2.3, 3.2 and 4.

*The OMI and GOME-2 tropospheric $NO_2$ algorithms are only mentioned in Section 2.3, without any detailed description, and conclusions are thrown as if they were consistent among them, while we know that large differences in tropospheric $NO_2$ exist between different product (applied on a same instrument) due to different algorithm hypothesis. The 2 algorithms should be described in Section 2.3 (maybe with a table of main differences) and impact of the algorithm hypothesis should be mentioned/quantified. Comparison e.g. of the model $NO_2$ profiles at the 3 stations type wrt to the profiles used as a-priori in the satellites retrievals would be interesting.*

Detailed descriptions of the OMI and GOME-2 tropospheric $NO_2$ algorithms have been added in section 2.3 and a table of the main features has been included. The possible impact of the two different algorithms on the study is mentioned in sections 3.2 and 4. However, it is not the purpose of this study to investigate the impact of the different algorithms. The model and satellite $NO_2$ a priori profiles have been described in sections 2.2 and 2.3.

More detailed answers are given in the specific comments below.

**Specific comments and technical corrections:**

*Page 1, line 9: consider changing "the main" into "one of the"*

The text has been revised accordingly.

*Page 1, line 13 (and in the conclusions): why using DOAS/MAX-DOAS while in the rest of the document only "MAX-DOAS" is used?*

The text has been revised accordingly.

*Page 1, line 18: consider reformulating "mainly due to the higher spatial resolution of OMI" to also mention the different products.*

The text has been revised accordingly.

*Page 2, line 22: consider changing "is limited… respectively" to "are respectively limited by the …"*
*Page 3, line 10: "are applied to satellite data …"* → *Only applied to OMI data!*

The text has been revised accordingly.

*Page 4, line 18: inconsistency between "a fixed azimuth angle of 255°" and "at azimuth angles of 80° relative to the solar azimuth". Reformulate.*

The text has been revised accordingly.

*Page 4, line 21: add reference for "only at the elevation angles of 15° and 30° in order to avoid uncertainties introduced due to aerosol loadings at lower elevation angles" and explain what is done with the VCD retrieved at the 2 elevation angles. Is the average value used? Or is a filtering related to the difference of the 2 VCD values used (as in Brinksma et al., 2008)?*

Initially, the average of measurements at both elevation angles had been used without any criteria. Following the above comment, an improved approach has been adopted for the revised manuscript. An average of measurements at both elevation angles is used when they agree within 20%, otherwise only results at 30° are compared with satellite retrievals. Text has been revised accordingly.

*Page 5, line 7: illustrate the shape of the used "mean vertical profiles from the air quality modelling tool consisting of the photochemical grid model CAMx and the mesoscale weather prediction system WRF" (ideally also showing how they differ from the 2 satellite $NO_2$ algorithms a-priori profiles)*

The model and satellite a priori profiles are similar to exponential decay with altitude. We describe them in the text (sections 2.2 and 2.3) but we have not included any figure in the manuscript. In the following figure the CAMx, OMI and GOME-2 a priori mean profiles are shown. All the a priori profiles are similar except for the CAMx profile of the urban site (UC), in which the $NO_2$ concentrations at the lowest layers are much higher due to the elevated NOx emissions in the urban area.

[Figure]

*Page 5, line 13: the 0.07 and 0.01 difference in AOD between the sites should be referred to the mean value. (0.07 wrt to 0.1 is large while it is small if compared to 1).*

This section has been revised. These differences have been calculated for AOD values ranging between 0.1 and 1.1. The long-term (1997-2005) mean AOD over Thessaloniki (UC) is 0.33±0.14 and 0.53±0.17 for winter and summertime, respectively.

*Why scaling the LIDAR profile to AOD values (line 14), if the "AOD loading and AOD profile between the three locations is very small" (line 12-13)? How much the choice of this aerosols a-priori profile affects the MAXDOAS NO2 VCD? Give an estimation of the MAXDOAS error.*

The same climatological LIDAR profile was used as a priori for all three locations. This profile is scaled during the simulations using AOD values in the range 0-1.5 only for the construction of the AMF LUT. The AMF corresponding to each measurement is extracted from the LUT using AOD measurements from the CIMEL sun-photometer operating in Thessaloniki. We have repeated the AMF simulations using an aerosol profile according to Shettle (1989) for autumn/winter conditions and the average difference in the retrieved $NO_2$ tropospheric VCDs was < 2%.

*Maybe add a subplot on figure 6 with the time-series of the Thessaloniki CIMEL AOD to have an idea of the variability?*

We think that this subplot is not necessary and it doesn't improve the discussion because we already take into account the variability in AOD for the $NO_2$ retrieval using the CIMEL AOD measurements as described above.

*Page 5, line 18: it's Fig 4 and not 3. Line 21 it's Fig 2 and not 4.*

The text has been corrected.

*Page 5, line 24-25: I would put this sentence at the end of the section.*

The text has been revised accordingly.

*Section 2.3: add a description of the 2 NO$_2$ algorithms maybe with an overview table) and references to papers (at least Valks et al., 2011 and Bucsela et al., 2013) and some other validation results (here on when discussing the results).*

The text has been revised accordingly. Section 2.3 has been completed with references, including the suggested, and a table for the NO$_2$ algorithms has been added. Also, validation results have been included in sections 2.3, 3.2 and 4.

*Page 6, line 17: "a particular direction and path" – how long is the representativeness area of the MAXDOAS? Give references (at least irie et al., 2011) and estimation ranges. In page 7, line 32, an area of 2x2km² is considered representative of the MAXDOAS data, but when using high elevation angles as 15° and 30°, and considering no aerosols, longer distances are obtained. How much the choice of using only one model cell of 2x2km² is affecting the adjustment factor?*

Sections 2.4 and 3.3 have been revised. We have estimated the representativeness area of the MAX-DOAS data based on the elevation angle and boundary layer height (BLH) reanalysis data from ECMWF (as the ratio of the BLH to the tangent of the elevation angle). An average of 0.55 km for both elevation angles was calculated as the representative horizontal distance of the MAX-DOAS measurements for the campaign period and OMI overpass time. Only ~2% of the campaign data were found to correspond to horizontal distances larger than 2 km. Thus, we believe that the 2x2km$^2$ model cell can be considered representative of the MAX-DOAS data. We also did calculations of the adjustment factors using more than one cell and they seem not to be significantly affected (Table 6 in the revised manuscript).

*Page 6, line 19: "the satellite data are adjusted" → only OMI data are adjusted.*

Text has been revised accordingly.

*Page 6, line 25 to page 7 line 24: a lot of details are given for the model data used here, which is too much compared to the lack of description of the satellite data in Sect 2.3. The goal of the paper is focusing on the satellite data!*

Section 2.3 has been revised.

*Page 7, line 25: again, only OMI data are adjusted.*

Text has been revised accordingly.

*Page 7, line 32: see comment above about the impact of using only one model cell of 2x2km² is affecting the adjustment factor.*

Answered in the previous comment about representativeness area.

*Page 7, line 33: "the method was applied only to OMI data because for GOME-2 the sub-satellite pixel is very large, covering typically the entire domain of Fig. 1" – see general comment: how the GOME-2 data would compare to an average of the 3 stations?*

The average NO$_2$ VCD values of the 3 stations are mostly affected by the large NO$_2$ loading of the urban area. Thus, the comparison of GOME-2 overpass data with the average of the 3 stations is similar to the comparison with the urban site observations. In the following figures the GOME2A (upper panels) and GOME2B (lower panels) overpass data are compared with the average of all three stations (left panels) and the urban site (UC) measurements (right panels).

[Figure]

*Page 8, line 14 and line 19: again, what columns are used in this study (in the figures, tables, etc)? VCD at 15° or at 30°? Or something else?*

Answered in a previous comment. This information is now included in the captions of figures and tables.

*Page 8, line 26: the "upper limit for the distance": does this mean that only the closest pixel within this distance is used for the satellite data? Or an average of the pixels values within this distance is used? How much choosing 50km for both instruments would affect the comparison?*

Both the pixel selection and the distance criterion are based on the satellite pixel size. In case of OMI the closest pixel within 25 km is used, in order to enhance the possibility that the station would be included in the pixel area, whereas for GOME-2 sensors, the pixels of which are much larger, an average within 50 km seemed to be a better choice. However, the comparison results are only slightly affected when we use a limit of 50 km also for OMI measurements. This is also the case if we choose the closest pixel of GOME-2 instead of the average value. Text has been revised.

*Page 8, line 29: typo: "table 3 and 4"*

The text has been corrected.

*Page 9, line 6: "can be attributed mainly to its smaller pixel size" and smaller distance criterium? "to its higher sensitivity in the boundary layer compared to GOME-2": give a reference. The fact that the 2 algorithms are different should also be mentioned and commented.*

The smaller distance limit does not affect the comparison results. There is a detailed answer in a previous comment. Section 3.2 has been revised accordingly and a reference for OMI high sensitivity in the

boundary layer has been added as suggested by the reviewer. The differences between the two algorithms are mentioned in the revised manuscript.

*Page 9, line 11: "The use of actual satellite geometry for each day is complex and would require a much larger domain for the air-quality simulations, more than double the currently used (120x120 km$^2$) in order to include all possible pixel sizes and positions for each location. However, such simulations were not available for this study."* → *This is really a pity. The study would be much more realist...*

We agree, but we didn't have such capability.

*Page 9, line 19: again, it would be nice to have a figure with the NO$_2$ model profile at the 3 sites.*

We have addressed this comment already and we have provided a figure. However, we have decided not to include this figure in the paper but discuss the shapes and give numbers of the scale height of the different profiles.

*Section 4: no discussion to previous validation results is included, neither in Section 3 or 4. This should be added. Discussion of the difference in GOME2A and GOME2B results should also be added. Again importance of the OMI and GOME2 algorithmic differences should be mentioned.*

The text has been revised accordingly. Validation results have been included and discussed in sections 2.3, 3.2 and 4. Discussion of the difference in GOME2A and GOME2B results has been included in sections 3.2 and 4. The OMI and GOME2 retrieval algorithms are described in section 2.3 and the importance of the algorithmic differences is mentioned.

*Page 18, table 2: mention that the statistics are for tropospheric NO$_2$. From which angle (12 or 30°)?*

The text and table caption have been revised accordingly.

*Page 18, table 3: discuss the larger differences in GOME2A wrt GOME2B mean values in RC sites and smaller in SC and UC (not consistent with the pixel size influence, as GOME2A is twice as small than GOME2B). A figure with location of the satellite pixels position/size could be useful.*

Sectioned 3.2 has been revised and examples of typical pixel sizes are now illustrated on the maps showing the mean NO$_2$ spatial distribution over the campaign period as observed by the satellite instruments (Fig. 5 in the revised manuscript).

*Page 19, table 4: add slope and intercept values to this table. Again, discuss GOME2A wrt GOME2B differences.*

Slope and intercept values have been added in the table (Table 5 in the revised manuscript). Discussion on GOME2A and GOME2B differences has been included in section 3.2.

*Page 19, table 5: considering adding a line with values over 2 or more cells in the MAXDOAS pointing direction (related to comment on the MAXDOAS representativeness).*

The comment about the representativeness area of the ground-based measurements has been addressed already. The adjustment factors calculated for more than one cells are presented in the table (Table 6 in the revised manuscript) and discussed in section 3.3.

*Page 20, figure 1: add a ruler to estimate distances.*

Ruler has been added in Fig. 1.

*Page 23, caption of figure 4: "and azimuth angle 100º relative to the sun for "➔ in the text (page 4, line 19) it's 80° !*

The Fig. 4 has been changed and the AMF lines are now referred to an azimuth of 80º relative to the sun.

*Page 24, figure 5: it would be good to add the locations of the 3 stations also on the 1rst panel, and add a ruler to estimate distances.*

Rulers have been added to both panels and the three stations are shown also on the 1st panel. Also, all CAMx cells have been now included in the 2nd map. Fig. 6 in the revised manuscript.

*Page 25, figure 6: why not including the period where the 3 instruments measured at the same site on the figure?*

A separate plot has been included in Fig. 2.

*Page 26, figure 7: use the same axis limit for the urban scatter plot for the 3 instruments.*

The axis limit has been adjusted accordingly (Fig. 8 in the revised manuscript).

*References:*

*Irie, H., Takashima, H., Kanaya, Y., Boersma, K. F., Gast, L., Wittrock, F., Brunner, D., Zhou, Y. and Van Roozendael, M.: Eight-component retrievals from ground-based MAX-DOAS observations, Atmos.*

*Meas. Tech., 4(1), 1027–1044, doi:10.5194/amtd-4-639-2011, 2011.*

*Bucsela, E. J., Krotkov, N. A., Celarier, E. A., Lamsal, L. N., Swartz, W. H., Bhartia, P. K., Boersma, K. F., Veefkind, J. P., Gleason, J. F., and Pickering, K. E.: A new stratospheric and tropospheric NO2 retrieval algorithm for nadir-viewing satellite instruments: applications to OMI, Atmos. Meas. Tech., 6, 2607– 2626.*

*Valks, P., Pinardi, G., Richter, A., Lambert, J.-C., Hao, N., Loyola, D., Van Roozendael, M. and Emmadi, S.: Operational total and tropospheric NO2 column retrieval for GOME-2, Atmos. Meas. Tech., 4, 1491– 1514, doi:doi:10.5194/amt-4-1491-2011, 2011.*

The references suggested by the reviewer have been included in the revised manuscript.

---

## Author Comment (AC2) · 23 Jan 2017

**Response to anonymous referee #2**

We would like to acknowledge the referee for their helpful and thorough review. We believe that their comments improved the quality of this work.

Some of the measurements and statistics presented in the study have changed as a result of some of the reviewers' suggestions. Moreover, an updated OMI overpass data set has been used, generated by the Aura Validation Data Center (AVDC) (http://avdc.gsfc.nasa.gov) in November 28th 2016.

At first, we address the reviewer's general comments remaining from the first review phase and then we answer to the reviewer's comments submitted during the interactive discussion. Our responses follow the reviewer's comments (in bold).

**General comments from first review phase:**

*Few satellite data and many different models are used but we can not find any map or plot from these data. Please try to visualize, what you use. I expect maps from all applied models and satellites (examples) in this study.*

Maps of NOx emissions, CAMx simulations and satellite observations (OMI, GOME2A and GOME2B) used in this study have been added (figures 5 and 6 in the revised manuscript).

*The three measurement areas (fig. 1) are more complex because of the sea, how is this problem (sea-land) in the model considered? This makes the selected albedo also complex.*

The land-use spatial distribution is accounted for in CAMx which considers 11 land-use categories namely: 1) Urban, 2) Agricultural, 3) Rangeland, 4) Deciduous forest, 5) Coniferous forest, wetland, 6) Mixed forest, 7) Water, 8) Barren land, 9) Non-forested wetlands, 10) Mixed agricultural/range, 11) Rocky (with low shrubs). CAMx assigns a single dominant land-use to each grid cell while translating the gridded land-use fields from the meteorological model WRF applied with the use of a 20-category MODIS-based land use database of 30 arc-second (~ 1 km) resolution. Each CAMx land-use type is associated with a single UV albedo, which can take one of the values 0.04, 0.05 or 0.08. Under this view, CAMx can distinguish between sea- and land-dominated grid cells and assigns the most appropriate albedo value.

*The selected areas overlap each other, how is this problem considered?*

We think that this is not a problem for our study. We investigate each location separately by averaging the modeled $NO_2$ tropospheric columns over a typical pixel size area.

*The campaigns have different time periods, to compare these data-sets, we need the same time period. Maybe an additional fit for the overlapping time!*

Unfortunately, for technical reasons we were unable to achieve a longer common period of measurements for the three locations which would increase the confidence of the results. The common period is too short for comparing ground-based with satellite data and deriving meaningful statistics. However, this study is mainly focused on comparing the ground-based data sets with satellite retrievals over locations characterized by different atmospheric pollution loadings. We do not compare directly the $NO_2$ observations at the three campaign sites. However, the NOx emissions and $NO_2$ levels observed in the

urban site are different compared to the other two sites during the whole campaign period, due to enhanced anthropogenic activities (mostly road transport). Thus, the differences in $NO_2$ loading are mentioned and discussed.

*Please clarify the wind directions for the period of the campaigns.*

We have investigated the wind data for the period of the campaigns. Certainly there is no prevailing wind direction that can be used to explain robustly differences of $NO_2$ concentrations among the three locations. One can identify cases where $NO_2$ from the city could be potentially transported towards the direction of the suburban site, but such attribution would be rather speculative since there is no other supporting information to make a consistent case.

**Comments during interactive discussion:**

*The map of satellite can show the variability of NOx and corresponding the emissions, this means, if the applied OMI data shows really the emitted from this area or not.*

Maps of the mean tropospheric $NO_2$ columns over the campaign time period measured by OMI, GOME2A and GOME2B have been included in the manuscript (figure 5 in the revised manuscript). It seems that OMI does not detect the elevated $NO_2$ concentrations of the urban area for this period of time.

*The comparison of the $NO_2$ tropospheric from different sites (UC, RC and SC) is less relevant because the data are from different time periods, for such comparison, you need the same time period.*

We absolutely agree. Therefore this study is mainly focused on comparing the ground-based data sets with satellite retrievals. There is no direct comparison of $NO_2$ data from the different campaign sites. However, the higher $NO_2$ levels observed in the urban area due to enhanced anthropogenic activities, which can be seen also from the CAMx simulations, are discussed.

*The adjustment factors are more depend on the model data than OMI data; this rescaling can be used but it is somehow banal. Without any adjustment factors, the results can be seen from fig. 4, what I expect for OMI reconstructed.*

We are not sure why the reviewer mentions Fig.4. In the discussion paper this figure shows examples of the AMFs calculated by radiative transfer modeling. The use of the adjustment factors calculated by model simulations seems to improve the comparison between the MAX-DOAS and satellite measurements over the urban area (figure 10 in the revised manuscript).

*Generally, you can compare every ground based with every satellite with different time periods but in this case, you can not compare the results together.*

In general we agree. However, the $NO_2$ spatial distribution in the greater area of Thessaloniki follows the NOx emissions pattern, which does not change significantly during the campaign period. In this study, we investigate how the satellite and ground-based data sets compare over locations characterized by different $NO_2$ concentration levels.

*P4, l16: direct sun light, we have only scattered sun light, but if we look directly in the sun (still scattered), the measured spectra have structures, which should be removed from the analysis.*

We do not use direct sun spectra in this study. We only mention the capability of the systems to perform direct sun measurements in the systems' description section.

*P5,l5: albedo: 0.1. The three different areas (UC, SC and RC) have definitely different albedos.*

Since uvspec is not a 3D radiative transfer model, we cannot use spatially variable albedo. Instead we assumed an average albedo over each area. An albedo of 0.1 can be considered representative of the urban site (UC) for the wavelength region 400-450 nm that is used in the DOAS analysis. For the other two locations (SC and RC) we have run the AMF simulations again using the more realistic value of 0.07 and the manuscript has been revised accordingly. However, the differences in tropospheric $NO_2$ VCDs are quite small and the comparison results are not significantly affected.

*Fig. 5 down: Some pixels are missing after averaging!*

Only the pixels included in the typical OMI pixel size area over each campaign site were available. We have extracted the rest of the pixels from the model simulations and we present the mean $NO_2$ spatial distribution for the whole domain in the revised manuscript (Fig. 6).

*Fig. 7: It is not clear to me, if you used all data from different campaigns with different time periods or not, if yes, they are not comparable in such form. You need a significant criteria, namely the same time period.*

Unfortunately, due to technical issues, the three instruments were performing measurements at the different sites in different time periods. However, this study focuses on the comparison between ground-based and satellite observations over areas with different $NO_2$ loadings and different spatial patterns. We do not compare directly the MAX-DOAS measurements at the different stations.

*Fig. 9: You can not compare the slopes from different time periods specially with seasonal difference!*

We agree that we cannot resolve with the available data any seasonal differences in the comparison of ground-based with satellite data. As argued already in other comments above, in this study we focus mainly on how different $NO_2$ levels and their spatial patterns affect the comparisons.

---

## Referee Report (RR1)

**Interactive comments on "Comparisons of ground-based tropospheric NO2 MAX-DOAS measurements to satellite observations with the aid of an air quality model over Thessaloniki area, Greece" by Theano Drosoglou.**

Referee:

In this manuscript, the authors report on a comparison of ground based tropospheric NO2 MAX-DOAS measurements to satellite observations with the benefit of an air quality model. The main point is the spatial resolutions between different instruments, for this purpose, different satellite data and model data were used.

I am going to comment only the author's responses.

I recommend the publication after at least these two revisions:

In this paper, we need some clarifications, at every point, if you want to compare the three selected areas with satellite data, you should add meaning of the time shift, for example Page 1, line 20; here it is important to know the time of the campaign SC area!!

The wind data during the campaign should be shown.

1. The main point in this study is the adjustment factors, which are not clear, how they are determined, please clarify this point, maybe by an example. Maybe by a rescaling of satellite data is this study easier to explain.

The additional explanation to this point is acceptable but it could be declared better.

2. Few satellite data and many different models are used but we cannot find any map or plot from these data. Please try to visualize, what you use. I expect maps from all applied models and satellites (examples) in this study.

Figures 5 and 6 are the answer to this question. From these figures, we can see the gradients from the satellites so that the selected areas have low NO2 values.

Actually, the figure 6 must be in the same frame as figures 5. The model data are used for the calculations of adjustment factors, it is interesting to see, if there is a strong or weak agreement between model and satellite data, this agreement should be done rough only from the maps.

3. The three measurement areas (fig. 1) are more complex because of the sea, how is this problem (sea land) in the model considered? This makes the selected albedo also complex.

Here, it was an easy answer, no, they did not.

4. The selected areas overlap each other, how is this problem considered?

Here it was the same answer, no. I think, at this point and last one, the autos neglect many important modeling points but I can understand that it is complicated and need the related knowledge.

5. The campaigns have different time periods, to compare these data-sets, we need the same time period. Maybe an additional fit for the overlapping time!

I think, the authors did not understand, how important the consistency of the data is. That is clear that you cannot repeat a campaign but this misunderstanding is represented in correlations and comparisons and consequently in the interpretations.

6. Please clarify the wind directions for the period of the campaigns.

Please add plots and data with references.

7. The map of satellite can show the variability of NOx and corresponding the emissions, this means, if the applied OMI data shows really the emitted from this area or not.

From the satellite maps, we can see that the selected areas have low values, for the transport by wind, we need the related wind.

8. The comparison of the NO2 tropospheric from different sites (UC, RC and SC) is less relevant because the data are from different time periods, for such comparison, you need the same time period.

9. The adjustment factors are more depend on the model data than OMI data; this rescaling can be used but it is somehow banal. Without any adjustment factors, the results can be seen from fig. 4, what I expect for OMI reconstructed.

Please see the answer number 1.

10. Generally, you can compare every ground based with every satellite with different time periods but in this case, you can not compare the results together.

Your measurements show few gradients but you speculate here that you do not have many gradients from satellite.

11. P4, l16: direct sun light, we have only scattered sun light, but if we look directly in the sun (still scattered), the measured spectra have structures, which should be removed from the analysis.

That is ok.

12. P5,l5: albedo: 0.1. The three different areas (UC, SC and RC) have definitely different albedos.

Albedo does not affect directly the NO2 VCDs, if you expect first to see different VCDs then adopt different albedos, you are wrong. You have different surfaces. That is clear that you will get different albedos but I understand that it is difficult for you.

13. Fig. 5 down: Some pixels are missing after averaging!

The answer is OK.

14. Fig. 7: It is not clear to me, if you used all data from different campaigns with different time periods or not, if yes, they are not comparable in such form. You need a significant criteria, namely the same time period.

I think, we discussed at different points about this issue, I do not want to repeat.

15. Fig. 9: You cannot compare the slopes from different time periods specially with seasonal difference!

These three correlations in a figure is an obvious sign that the areas will be compared together.

By such a time shift of ground based measurements, it does not make much sense, the shape and the gradients of OMI is more and less the same for all three areas, now, it is important, at which time period, you did the measurements, for example SC area, that means, if the data are close to the background, the correlation is better. For this reason, you should remove these correlations or add enough clarification.

---

## Author Response (AR2)

We would like to acknowledge the editor and the referees for their helpful and thorough review.

We have revised the manuscript according to editor's suggestions and the second reviewer's comments. Our responses (in blue) follow the editor's and reviewer's comments (in black italics).

**Response to editor's suggestions**

*In page 13, line 22-24, you state that the use of actual satellite geometry for each day is complex and would require a much larger domain for the air-quality simulations, more than double the currently used (120 km×120 km) in order to include all possible pixel sizes and positions for each location. It is difficult for me to understand this statement as you have mentioned (in page 11, line 26-29) that the OMI sensor provides better detection of higher local pollution levels [Kramer et al., 2008], due to its higher spatial resolution (13 km × 24 km at nadir), compared to GOME-2 sensors which detect the average NO2 concentration over a larger area (40 km × 40 km for MetOpA and 80 km × 40 km for MetOpB). Should some outmost pixels at the swath edges be discarded or not when making a comparison with model simulations?*

We have excluded some outmost OMI pixels from our study, using the cross-track position value and, as a result, all the remaining pixels are well within the domain of the CAMx simulations. Thus, we were able to re-calculate the adjustment factors using a close-to-actual pixel size and position, which led to a significant improvement in the comparison results. Several parts of the manuscript have been revised to account for the new calculations. The main revisions are listed below.

- In sections 2.4 and 3.3 the description of the calculation of the adjustment factors and the reconstruction of OMI have been revised.
- In sections 3.2 and 3.3 the comparison and some parts of the discussion have been revised accordingly.
- Revisions have been made in Tables 4, 5 and 6.
- Figures 8, 9 and 10 have been revised (Fig. 10 has been separated in Fig. 11 and Fig.12).
- The results mentioned in the abstract and the section of the conclusions have been revised accordingly.

*In the last paragraph of Sect. 3.3, you may add a short discussion of the aerosol shielding effect on the comparison between MAX-DOAS and satellite observations. There have been some papers addressing this issue, e.g.,*

*Ma, J. Z., Beirle, S., Jin, J. L., Shaiganfar, R., Yan, P., and Wagner, T.: Tropospheric NO2 vertical column densities over Beijing: results of the first three years of ground-based MAX-DOAS measurements (2008-2011) and satellite validation, Atmos. Chem. Phys., 13, 1547-1567, 10.5194/acp-13-1547-2013, 2013.*

*Jin, J., Ma, J., Lin, W., Zhao, H., Shaiganfar, R., Beirle, S., and Wagner, T: MAX-DOAS measurements and satellite validation of tropospheric NO2 and SO2 vertical column densities at a rural site of North China, Atmospheric Environment, 133, 12-25, http://dx.doi.org/10.1016/j.atmosenv.2016.03.031, 2016.*

A short discussion on the aerosol shielding effect has been included in the end of section 3.3. The suggested references have been also included.

*Please also consider the following technical issues when making a revision of your manuscript.*

*Page 1, line 22: Change "resulted to" to "resulted in".*

*Page 1, line 23: Change "Phaethon" to "MAX-DOAS".*

*Page 2, line 29: Change "date back in" to "date back to".*

*Page 3, line 31: what is the meaning of OH here?*

*Page 4, line 14 and line 16: Change "has been operating" to "was operated".*

*Page 4, line 33, "Fig. 2": Please renumber the figure in Page 30 (change "Figure. 3" to "Figure 2") and move it to the previous page, just after Figure 1.*

*Page 5, line 28, "Fig. 4": Fig. 3 has not been mentioned before. Please change "Fig. 4" here to "Fig. 3", renumber the figure in Page 31 (change Figure 4" to "Figure 3") and move it to the previous page, just after the updated Figure 2.*

*Page 5, line 31, "Fig. 2": Please change "Fig. 2" here to "Fig. 4", renumber the figure in Page 29 (change Figure 2" to "Figure 4") and move it to the back page, just after the updated Figure 3.*

*Page 5, line 31,Table 2: "Table 1" should have appeared somewhere in the text before.*

*Page 14, line 7: Change "all three instruments" to "all three satellite instruments".*

All the above technical issues have been corrected in the revised manuscript.

**Response to the comments of Anonymous referee #2**

*In this paper, we need some clarifications, at every point, if you want to compare the three selected areas with satellite data, you should add meaning of the time shift, for example Page 1, line 20; here it is important to know the time of the campaign SC area!!*

The $NO_2$ time series over the UC site has been separated into two subsets which coincide with the different periods of measurements at the RC and SC sites, i.e. from 1 November 2014 through 31 January 2015 and from 20 January through May 2015 respectively. Revisions have been made in Sect. 3.2 and 3.3 of the manuscript and the conclusions. Also, Fig. 8 has been revised and Fig. 10 (previous version of the manuscript) has been separated into two different figures (Fig. 11 and Fig, 12).

*The wind data during the campaign should be shown.*

The wind data are shown in Fig. 10 and a short discussion has been included in Sect. 3.3.

[revised manuscript text omitted]